# Assimilating CryoSat-2 freeboard to improve Arctic sea ice thickness estimates

Imke Sievers[1,3], Till A. S. Rasmussen[1], and Lars Stenseng[2]

[1]Danish Meteorological Institut, Lyngbyvej 100, Copenhagen East, Denmark
[2]DTU Space, Danish Technical University, Elektrovej Bygning 328, 2800 Kongens Lyngby, Denmark
[3]Aalborg University, A. C. Meyers Vænge 15, 2450 Copenhagen, Denmark

**Correspondence:** Imke Sievers (imksie@dmi.dk)

**Abstract.** In this study, a new method to assimilate satellite radar altimetry derived freeboard (FB) is presented with the goal of improving the initial state of sea ice thickness predictions in the Arctic. In order to quantify the improvement in sea ice thickness gained by assimilating FB, we compare three different model runs. One reference run (refRun), one that assimilates only sea ice concentration (SIC) (sicRun) and one that assimilates both SIC and FB (fbRun). It is shown that, estimates for both SIC and FB can be improved by assimilation, but only the fbRun improved the FB. The resulting sea ice thickness is evaluated by comparing sea ice draft measurements from the Beaufort Gyre Exploration Project (BGEP) and sea ice thickness measurements from 19 ice mass balance buoys (IMB) deployed during the Multidisciplinary drifting Observatory for the Study of Arctic Climate (MOSAiC) expedition. fbRun's sea ice thickness compares better to the longer BGEP observations and poorer than refRun and sicRun to the shorter MOSAiC observations. Further, the three model runs are compared to the Alfred Wegener Institute's (AWI) weekly CryoSat-2 sea ice thickness, which is based on the same FB observations as were assimilated in this study. It is shown that, the FB and sea ice thickness from the fbRun is closer to the AWI CryoSat-2 values than the ones from refRun or sicRun. Finally, comparisons of the above-mentioned observations and both the fbRun sea ice thickness and the AWI weekly CryoSat-2 sea ice thickness were performed. At the BGEP locations, both fbRun and the AWI CryoSat-2 sea ice thickness perform equally. The total root-mean-square error (RMSE) at the BGEP locations equal 30 cm for both sea ice thickness products. At the MOSAiC locations the At the MOSAiC locations, fbRun's sea ice thickness performs significantly better, with a total 11 cm lower RMSE.

## 1  Introduction

With declining sea ice in the Arctic, marine traffic is increasing (Cao et al., 2022). This increases the demand for accurate sea ice predictions to ensure safety on the routes. Data assimilation is a commonly used tool to improve the initial state of sea ice predictions (Chen et al., 2017; Mu et al., 2018; Fiedler et al., 2022). In data assimilation, models and observations are combined using a number of approaches. For all approaches, the variables that are assimilated need to be observable and need to affect the model variable that the assimilation aims to improve. Stroeve and Notz (2015) lists sea ice volume and ocean heat content as the two model variables with the largest impact on Arctic sea ice forecast. Ocean heat content is difficult to observe on an Arctic wide scale, but sea ice concentration (SIC) and sea ice thickness can be observed from satellites (Kwok, 2010;

Laxon et al., 2013; Ivanova et al., 2014; OSISAF, 2017; Hendricks et al., 2021). While satellite-observed SIC has rather good accuracy and has been available since the late 1970s, satellite sea ice thickness observations have only been available since the early 2000s and come with large uncertainties (Laxon et al., 2003; Kwok, 2010). Several studies have found that sea ice thickness, in contrast to SIC, has a longer memory (Day et al., 2014; Stroeve and Notz, 2015; Dirkson et al., 2017). Longer memory here means that the change introduced by initial sea ice thickness persists longer than change introduced by SIC. This makes the sea ice thickness the more suitable variable to assimilate when aiming for an improved initial estimate of the Arctic sea ice, which also has an impact on the skill of the forecast at longer timescales (Day et al., 2014).

Arctic wide sea ice observations can only be obtained through remotely sensed data from satellites. However, for sea ice thickness, it is possible to observe the portion of the sea ice above the sea surface, which is referred to as freeboard (FB). The longest FB observations from a satellite with a polar orbit are obtained from the European Space Agency (ESA) satellite CryoSat-2, which has been in orbit since 2010 (Drinkwater et al., 2004). Using an advanced radar altimeter, data from CryoSat-2 can be used to estimate FB as the difference between the observed height of the sea ice surface and the water level in leads between sea ice floes. To derive sea ice thickness from FB, a number of assumptions need to be made, which will be discussed below. These assumptions lead to a large uncertainty in the resulting sea ice thickness estimate. Therefore, we propose a method that assimilates FB directly, instead of sea ice thickness derived from FB.

Most existing sea ice thickness products use FB measurements to calculate sea ice thickness under the assumption of hydrostatic balance. The hydrostatic balance equation relates sea ice thickness to FB, snow density, snow thickness, sea ice density, and seawater density. In this relation, FB is measured, and the other parameters are derived from climatologies or empirical values derived from in situ observations (Ricker et al., 2014; Kwok and Cunningham, 2015; Tilling et al., 2018). The above-mentioned uncertainties in satellite derived sea ice thickness largely originate from the uncertainty of these parameters (Alexandrov et al., 2010). According to Alexandrov et al. (2010), sea ice density introduces the largest error when calculating sea ice thickness from FB under the assumption of hydrostatic balance. Sea ice density depends on the ice age, where younger sea ice has a higher salinity due to more brine being enclosed in it. Over time, brine is expelled into the ocean below. During the melt season, salt is washed out by meltwater (Cox and Weeks, 1974), making multi-year ice (MYI) less saline and therefore less dense than first year ice (FYI). Enclosed gas is another parameter that makes sea ice density estimates uncertain. FYI sea ice density uncertainty is typically around 23.0 $kg/m^3$, and for MYI, the uncertainty is around 35.7 $kg/m^3$ (Alexandrov et al., 2010). This high uncertainty originates from the difficulty of measuring sea ice density and the limited availability of density measurements. The density varies within the ice column depending on whether the ice is below or above sea level. On top of that, the harsh environment adds extra challenges in performing exact measurements (Timco and Frederking, 1996). Despite the variation in sea ice density, most products use fixed values of 917 $kg/m^3$ for FYI and 882 $kg/m^3$ for MYI (Sallila et al., 2019). The second-largest error contributor to sea ice thickness, according to Alexandrov et al. (2010), is FB. Uncertainties in FB originate from uncertainties in the sea surface height, the location of the backscattering horizon, speckle noise (Ricker et al., 2014), the retracking of the radar waveform (Landy et al., 2019) and uncertainties in snow height and density used to calculate the reduction in radar wave propagation speed in the snowpack (Mallett et al., 2020). The uncertainty introduced by the snow thickness is heavily discussed (Kurtz and Farrell, 2011; Kwok et al., 2011; Laxon et al., 2013; Kern et al., 2015; Gar-

nier et al., 2021). Historically, snow thickness has been derived from the Warren et al. (1999) snow climatology (W99), which is calculated from Russian drift stations during the period 1954–1991. Most of the included measurements were obtained on thick MYI. However, Kurtz and Farrell (2011) showed that W99 is less reliable over FYI compared to MYI, and Laxon et al. (2013) proposed a method to differentiate MYI and FYI snow thickness and snow density from W99. This method is now more commonly used in sea ice thickness products than the pure W99 climatology (Sallila et al., 2019). Another alternative to W99 is to use a snow model to calculate the local snow thickness, depending on precipitation. For example, Fiedler et al. (2022) showed results using snow thickness from the global coupled sea ice ocean model Forecast Ocean Assimilation Model (FOAM (Blockley et al., 2014)) or Landy et al. (2022) using the SnowModel-LG (Liston et al., 2020).

W99 also includes a snow density climatology, which was commonly used in the calculation of sea ice thickness until 2020 (Sallila et al., 2019). Mallett et al. (2020) found that approximating the snow density by a linear function improves the sea ice thickness estimate by about 10 cm. Recent sea ice thickness products, as for example Hendricks et al. (2021), have started to use the proposed seasonal linear approximation of snow density with good results. Seawater density only varies very little throughout the Arctic. Most CryoSat-2 sea ice thickness products use a single value of 1024 $kg/m^3$, which is the density at the freezing point of Arctic surface water. The influence of the uncertainty of this value on the hydrostatic balance equation is negligible (Kurtz et al., 2013).

The uncertainties in sea ice density, freeboard (FB), snow density, and seawater density all contribute to the overall error in sea ice thickness calculated from FB. To account for these errors, error estimates are used in data assimilation methods such as Kalman filters. Kalman filters rely on knowledge of the model uncertainties and observational uncertainties, as well as the assumption that they are unbiased and Gaussian distributed. Based on these assumptions, the Kalman filter aims to derive the best estimate. The accuracy of the resulting state estimate improves with a better uncertainty estimates. The errors in CryoSat-2-derived sea ice thickness are not only due to the sources mentioned above, but also depend on how FYI and MYI are defined. The sea ice density, snow thickness, and, in some cases, snow density are calculated based on this ice type. The ice type is typically derived from the OSISAF ice type data (Sallila et al., 2019), which distinguishes between FYI, MYI and ambiguous ice type (Aaboe et al., 2021). Ye et al. (2023) assessed different sea ice type products, including the OSISAF ice type data product, and compared it to the NSIDC sea ice age data (Tschudi et al., 2020). They found that the OSISAF ice type data for FYI has a bias of $0.42 - 0.6 * 10^6$ km$^2$ and for MYI of $-0.54 - -0.35 * 10^6$ km$^2$. This comparison only considers FYI and MYI areas and compares them to satellite obtained ice age products. Ambiguous areas are not considered. In most CryoSat-2 sea ice thickness products, a small transitioning area is assumed where a linear transition from MYI to FYI is assumed (Laxon et al., 2013; Tilling et al., 2018; Hendricks et al., 2021). However, the ice chart based sea ice type data product G10033 (Fetterer and Stewart, 2020) suggests large areas of mixed ice types. This area is notably larger and less homogeneous than the area suggested by the linear transition between MYI and FYI based on the OSISAF sea ice type. This means that sea ice density, snow thickness, and snow density errors are systematically underestimated or overestimated in this area of ambiguous ice type.

As the FB error estimate is part of the sea ice thickness error estimate, it is fair to conclude that the FB error is better constrained than the sea ice thickness error. This is not to say that FB errors are unbiased. However, by choosing to assimilate FB, error contributions originating from snow thickness, snow density, sea ice density, and sea ice type when converting FB

to sea ice thickness are eliminated. Consequently, it follows that the FB data would be more suitable for assimilation than the derived sea ice thickness, as a lower uncertainty will increase the weight of the observed CryoSat-2 FB.

The challenge of this approach is that FB is not a sea ice model state variable, but a diagnostic variable. Even though FB is not a state variable, it is related to sea ice thickness, which is a state variable, and can be calculated from FB under the assumption that a change in FB is caused only by modelled sea ice thickness and modelled snow thickness and that snow density and ice density are realistic.

In this study, we present an approach to assimilate FB directly into the sea ice model CICE (Hunke et al., 2017). We aim to answer the questions: "Does FB assimilation have a significant impact on the modeled sea ice thickness?" and "How does the modeled sea ice thickness after assimilation of FB compare to SIT from a conventional CryoSat-2 sea ice thickness product?" To transform FB into the model state variable sea ice thickness, we use parametrization and assumptions from the model and the forcing data. The method is implemented into CICE, but should be applicable to any other model. This study mainly focuses on CryoSat-2 measurements, but the approach presented could also be applied to ICESat FB data (Martino et al., 2019) with small adjustments. Several studies have mentioned approaches to assimilate FB (Vernieres et al., 2016; Kaminski et al., 2018; Fiedler et al., 2022), but none included a description of how the FB assimilation was implemented. Kaminski et al. (2018) conducted a study using the quantitative network design approach to quantify how beneficial it would be to assimilate radar FB, among other variables. The study concludes that assimilation of radar FB can improve sea ice volume simulations on the same order of magnitude as sea ice thickness assimilation. The quantitative network design approach builds upon error propagation and the sea ice thickness errors used in the analysis, which originate from the AWI CryoSat-2 sea ice thickness products. As discussed above, this error estimate includes no contribution from ice type data and might be underestimated. To our knowledge, this is the first paper presenting detailed descriptions of an assimilation method using FB instead of sea ice thickness.

## 2 Methods and data

The following section presents all data sets, software and methods used to derive the sea ice thickness data sets evaluated in this study. The model set up is presented in section 2.1, the assimilation set up is presented in section 2.2, the observational data are presented in section 2.3 and 2.4, and section 2.5 presents the observation data sets which are used for validation.

### 2.1 Model set-up

The FB assimilation is implemented in a coupled sea ice (CICE v6.2, Hunke et al. (2021b)) and ocean model (NEMO v4.0, Madec et al. (2017)). The coupling is based on Smith et al. (2021), however both NEMO and CICE have been updated to more recent versions. NEMO is set up following (Hordoir et al., 2022).

CICE is a multicategory sea ice model that consist of a dynamical solver, an advection scheme, and a thermodynamic column physics model called Icepack. CICE and Icepack (Hunke et al., 2021a) are developed independently, but are by default linked (Hunke et al., 2021a, b). The model is run with 5 thickness categories with category bounds that follow a WMO standard setup. The upper bounds for the 5 categories (n) are: n=1: 0.3 m, n=2: 0.7 m, n=3: 1.2 m, n=4: 2 m, n=5: 999 m. In the presented

study, CICE was implemented close to the default setup except that form drag calculations, following Tsamados et al. (2014), were enabled.

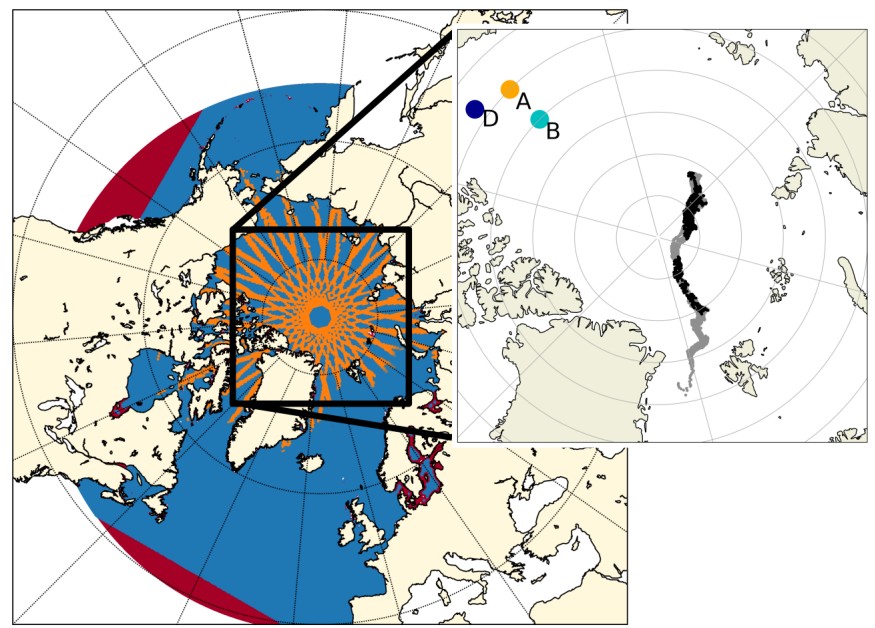

**Figure 1.** The red area indicates the model domain (large parts are covered by the blue and orange visualization) described in section 2.1, the blue area shows the OSISAF SIC data coverage and the orange lines give an example coverage of one week of CryoSat-2 data (here 2020-03-30). The zoomed area shows the location of the three moorings described in section 2.5 marked with according letters and the gray and black track indicates the drift path of the ice mass balance buoys also described in section 2.5. The gray line indicates the full data set used in figure 8 and the black subset the data set used in figure 7.

The model domain is pan Arctic, as shown by the red area in figure 1 (large parts are covered by the blue and orange visual-
130 ization). The lateral boundaries are located outside the Arctic sea ice covered region, such that sea ice boundary conditions are not required. The lateral ocean boundaries are forced with monthly GLORYS12 data, which consist of salinity, temperature, u- and v-velocities (Lellouche et al., 2021). The ocean model includes tides and the tidal forcing at the open boundaries originates from the TPXO 7.2 harmonic tidal constituents (Egbert and Erofeeva, 2002) and river runoff is based on a climatology from Dai and Trenberth (2002). The model is forced with 3 hourly ERA5 atmospheric forcing data, which consist of 2-m temper-
135 ature, 2-m specific humidity, 10-m wind, incoming shortwave and long wave radiation, total precipitation, snowfall and air pressure at sea level (Hersbach et al., 2017). The model runs discussed in this study are restarted from the same initial run, which run from 1995 to 2020 and was initialized from ORAS5 (Zuo et al., 2019) ocean temperature and salinity fields. The year 2010-2020 of the initial run were used to calculate the model background error discussed in section 2.2. The three other runs discussed in the following text are the refRun, sicRun and fbRun. RefRun consists of the initial run from 01-01-2018 to
140 31-12-2020. SicRun and fbRun are started from the same restart file as RefRun on 01-01-2018, but assimilate SIC and SIC

and FB respectively. They both also cover the period 01-01-2018 to 31-12-2020. All model output discussed in the following sections is calculated based on daily means.

In order to be able to assimilate radar FB from CryoSat-2 a new variable for radar FB needs to be introduced in CICE. For this we combined equation (4) from Alexandrov et al. (2010) with equation (12) from Tilling et al. (2018) to:

$$FBr = \frac{hi(rho_w - rho_i) - rho_s h_s}{rho_w} - (h_s(\frac{c}{cs} - 1.) \tag{1}$$

$hi$ is the modelled sea ice thickness from CICE, $rho_w$ is the modelled surface water density from NEMO, $h_s$ the modelled snow thickness from CICE, $c$ is the speed of light in vacuum ($3 * 10^8 m/s$) and $cs$ the speed of light in snow. $cs$ is calculated following equation 2:

$$cs = c(1 + 0.51 rho_s)^{-1.5} \tag{2}$$

Mallett et al. (2020) compared constant $rho_s$ values to the Warren et al. (1999) derived seasonal linear variation of $rho_s$ and concluded that a seasonal varying $rho_s$ can improve FB derived sea ice thickness estimates by up to 10 cm. The original value used in CICE is constant and equals 330 $kg/m^3$. In this study it was substituted with the derived relation from Mallett et al. (2020) follows equation 3:

$$rho_s = 6.5 * t + 274.51 \tag{3}$$

where t is time counted in month since October. The relation in equation 3 is only used in the radar FB calculation for the assimilation and nowhere else in the sea ice model. CICE uses constant $rho_i$ values, but for the radar FB calculation a variable sea ice density was needed since $rho_i$ has significant impact on equation 1 (Alexandrov et al., 2010; Kern et al., 2015). Sea ice density is dependent on the air bubbles enclosed in the sea ice and on the brine content (Timco and Frederking, 1996). Brine content in sea ice results from the brine rejection during freeze-up and drains over time. If the brine channels are not filled with water, they remain as air bubbles in the ice (Timco and Frederking, 1996). CICE calculates the salinity content in sea ice and the density of sea ice without accounting for a changing amount of air pockets. To calculate the sea ice density we divide the sea ice volume in one grid cell into fresh ice and brine, calculate the percentage of the fresh ice and brine and weight a fresh ice density ($rho_{i0}$) and the brine density ($rho_b$) with this.

$$rho_i = aice_b * rho_b + (1 - aice_b) * rho_{i0} \tag{4}$$

$aice_b$ is the amount of brine in percentage of the total ice volume. $rho_{i0}$ was set to 882 $kg/m^3$ following Alexandrov et al. (2010) values for MYI sea ice density. In the following text, FB stands for the radar FB.

## 2.2 Assimilation set up

Kalman filter-based assimilation is a widely used technique that employs an ensemble of model forecasts to estimate the state of a system using available observations. The method involves three main steps: a forecasting step, a filtering step and a re-sampling step. The forecast is performed by the model. During the filtering step, the ensemble members are adjusted based

on the knowledge of the model background error, observation error, model states, and observations to obtain the best possible estimate of the system state. In the re-sampling step, the best estimate from the filtering step is used to update the ensemble members. This process is repeated iteratively in order to improve the accuracy of the state estimate. For the filtering step, we use the Local Error Subspace Transform Kalman Filter (LESTKF) (Nerger et al., 2012), which is included in the Parallel Data Assimilation Framework (PDAF) (Nerger and Hiller, 2013). The LESTKF has prior to this study successfully been used to assimilate SIC and sea ice thickness by, for example Chen et al. (2017). In this study, PDAF is used offline, which means that the assimilation scheme runs independently of the ocean and sea ice model. The consequence is that the ocean and sea ice model needs to be restarted when the model and the assimilation exchanges information. PDAF was run separately for SIC and FB. Figure 2 illustrates the data flow between the different components. The numbers noted in the lower corner of each component corresponds to each of the following sections, describing which part of the assimilation is handled in which program.

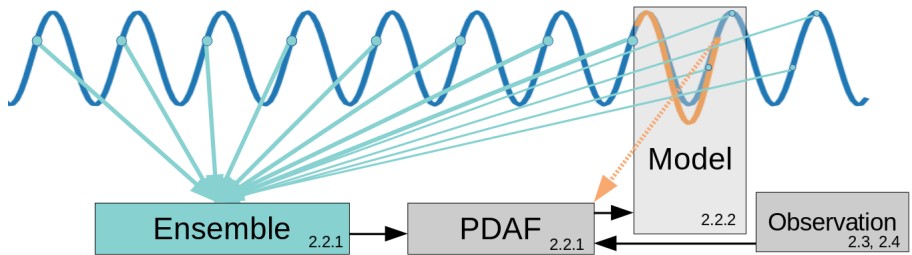

**Figure 2.** General set up of the assimilation routine. The dark blue curve indicates the initial model run and the orange curve the assimilated run, with the dashed orange arrow indicating the model state at the assimilation time. The turquoise thick error indicate the 8 days chosen around the assimilation date, and the thin turquoise arrows the 4 (or 3) days chosen plus minus 2 months around the assimilation date (described in section 2.2.1). The numbers in the lower corners indicate in which section of the paper the different elements are described.

### 2.2.1 PDAF

PDAF input consist of the model state, model ensemble, the observations and observation uncertainties on the model grid. The spread of the ensemble is used to calculate the model background error used in the filtering step. In this study, we only run one model realization and calculate the model background error in the Kalman filter from a static ensemble, similar as done by the set-ups in BALMFC (Nord et al., 2021) and SAM-2 (Tranchant et al., 2006). Using a static ensemble has the advantage of lower computational cost. To calculate the model background error based on a static ensemble, a free model run of the model used in the assimilation is needed. In our case the free model ran from 1995 to 2020, but only the years 2010-2020 were used to construct the static ensemble as the earlier years were considered to be spun up. The justification of using a static ensemble is based on the assumption that the model error at a certain day in a year is reflected by the inter annual model variability of this same day. Knowing the biases of the model allows for correction to this assumption. In our case, the model overestimates the ice extent, which we found when comparing the 10 years initial run to OSISAF (Saldo, 2022) SIC observations. Thus, the

background error based on the same date in several years would not result in a large enough spread to weight the observations correctly. The ensemble used to calculate the model background error consists of 80 members, and it is constructed as follows: Each of the 10 years from the free run is contributing with 8 days.

- In 8 years, 8 consecutive days are chosen starting from the date 3 days prior to 4 days past the assimilation time step.

- In 2 years, 3 consecutive days from the date 2 months prior to the assimilation time and 4 from 2 months past the assimilation time are chosen.

After the ensemble members were chosen, they are averaged. This average is then subtracted from each member, and the resulting variation is added to the model state at the assimilation date. These 80 ensemble members are then used to calculate the model error. For the observation error, we use the error estimates provided in the data sets.

### 2.2.2 Integration of increments

The physical model in section 2.1 utilizes the Kalman filter increment, which is the correction that adjusts the model state to the optimal state based on observations and model states. This increment is obtained as the difference between the model state input to PDAF and the analyzed state. The model state is corrected towards the analyzed state by subtracting the increment from the model state. To ensure stability, the increment is divided by the number of time steps (number of model time steps in one assimilation time step), which results in the fractal increment or the amount of change needed per model time step (following equation 5). This fractal increment is hereafter subtracted at each time step from the model value. This method is called incremental analysis updating and was introduced by Bloom et al. (1996). For SIC, this method is straight forward, since the observations are also what we aim to assimilate.

$$inc = \frac{var_0 - new_{ice}}{time_r} \tag{5}$$

FB needs to be converted into sea ice thickness, and if this would be done separately at each time step, the changing sea ice density and snow thickness could potentially influence the resulting sea ice thickness. Similar to SIC, the FB increment is subtracted from the model state at $t_0$. To convert FB to sea ice thickness, equation 1 was rewritten to:

$$new_{ice} = \frac{rho_s h_s + rho_w(FB_{new} + corr)}{rho_w - rho_i} \tag{6}$$

$new_{ice}$ is now subtracted from the modelled sea ice thickness and linearly spread following 5.

At each time step, we have the fractional increment of SIC and sea ice thickness to be subtracted from the model state. The model used in this study is a multicategory model. Therefore, the grid cell average increment must be spread over the five model categories. To achieve this, equation 7 was used. Here $var_{old}$ is the SIC at the current time step, $var_{old}(n)$ is the SIC in the n categories, $inc$ is the SIC increment and $n$ is the thickness category.

$$var(n) = var_{old}(n) - var_{old}(n)\frac{inc}{var_{old}} \tag{7}$$

In case the where SIC and FB are negative after the assimilation, they are rounded to 0. In cases where the SIC ends up above 1, SIC is rounded to 1. FB is only assimilated if SIC is above 80% and if sea ice thickness is above 0.05 m. These thresholds were chosen both for stability, but also because thin FB is not measured accurately (Wingham et al., 2006; Ricker et al., 2014) and because FB is calculated from the model's ice volume per unit area of ice. In areas with lower concentrations, this can lead to SIT and FB values that are unrealistically high. To ovoid over estimation of FB following this artifact, a high SIC threshold was chosen for the FB assimilation.

## 2.3    CryoSat-2 radar altimetry freeboard and sea ice thickness

The observed FB assimilated in this study is level 3 weekly gridded CryoSat-2 radar FB downloaded from the Alfred Wegener Institutes (AWI) sea ice portal (version 2.4 (Hendricks et al., 2021)). It is gridded, along track data on the EASE2-Grid with a 25 km resolution. The radar FB is defined as the elevation of retracked point above instantaneous sea surface height without snow range correction. The data product is derived from the CryoSat-2 baseline E data, the mean sea surface model DTU21, and the "Threshold First Maximum Retracker Algorithm" (TFMRA) (Ricker et al., 2014).

With the onset of melt in the beginning of summer, melt ponds are formed on the sea ice surface. The radar signature from melt ponds is comparable to the signature from leads, which can result in ambiguous determination of the sea surface height. This ambiguity results in a larger bias in the FB measurements, and FB data are therefore only assimilated from November to March, where we do not expect melt ponds. The uncertainty of FB given in the AWI data set ranges on average from 0 to 0.07 m in the chosen month. The data set was biliary interpolated to the model grid with help of CDO (Schulzweida, 2022). An example of the FB data assimilated per one assimilation time step (one week) is indicated by the orange lines in figure 1.

The data set also contains sea ice thickness derived by assuming hydrostatic balance, which is the method referred to as the classical approach. In order to obtain sea ice thickness from FB, hydrostatic balance is assumed, and sea ice thickness is calculated as described in equation 6. In the AWI CryoSat-2 data set, the snow thickness from Warren et al. (1999) snow climatology was applied over MYI and NSIDCs AMSR2 snow depth (Hendricks et al., 2021) was applied over FYI. The snow density is calculated following equation 3 from, Mallett et al. (2020) and the sea ice density is set to $916.7 \ kg/m^3$ for FYI and to $882.0 \ kg/m^3$ for MYI. MYI and FYI is distinguished with the help of OSISAF ice type data. For a more detailed description of the data set, see Hendricks et al. (2021).

## 2.4    OSISAF data

Ocean and Sea Ice Satellite Application Facility (OSISAF) SIC is assimilated in this study. It is based on the Special Sensor Microwave Imager / Sounder (SSMIS) passive microwave measurements, which is onboard a polar orbiting satellite. The OSISAF algorithm combines SSMIS microwave measurements with numerical weather prediction (NWP) model output from ECMWF in order to calculate SIC. Passive microwave measurements are independent of visible light, which makes this sensor type especially suitable in polar regions. The data set used is the climate data record (CDR) OSI-430-a which is gridded on a 25x25 km grid once a day. The data can be downloaded from the Norwegian Meteorological Institute FTP severs. The presented data set was chosen after examining the error estimates in the different data products. The comparison showed that the CDR

is the only data set that has no large error fluctuations over open water areas. More details on the error estimate can be found
in (Saldo, 2022). Studies have found that the summer melt ponds lead to underestimated SIC in satellite passive microwave
measurements (Kern et al., 2016; Ivanova et al., 2013; Rösel and Kaleschke, 2012). This is the reason we decided to only
assimilate SIC during the month November to March.

For the assimilation, the data set was bi-linearly interpolated on to the model grid using CDO (Schulzweida, 2022). The
resulting SIC data coverage assimilated is indicated by the blue area in figure 1.

## 2.5 Validation data

Two in situ sea ice observation data sets are used for validation. The Beaufort Gyre Exploration Project (BGEP) upward
looking sonar (ULS) sea ice draft data set and 19 ice mass balance (IMB) buoy deployed during the Multidisciplinary drifting
Observatory for the Study of Arctic Climate (MOSAiC) campaigned measuring sea ice thickness. The advantage of these
observations is that they are independent of the assimilated data, however each observation has limitations in terms of time and
space.

The BGEP ULS sea ice draft data set can be downloaded from www2.whoi.edu. The ULS data are obtained from three
locations named mooring A, B and D, marked with orange, turquoise and dark blue dots in figure 1. The data covers 2 years,
from October 2018 to November 2020. The instruments are located 50-85 m below the water surface and measure the ice
draft with a frequency of 2 seconds over a 2x2 m area. The signal is filtered and averaged over 10 second intervals in order to
correct for tilting errors. Tilting error refers to the error that results from the movement of the ULS when ocean currents move
the instrument and so influence the distance to the sea ice. The error is assumed to be random, hence averaging the data will
eliminate it. The sea ice draft accuracy is $\pm 5$ cm.

For the comparison of BGEP observations and model and AWI data, the model and AWI draft was calculated as sea ice
thickness minus sea ice FB. To compare the BGEP data with the three model runs, the daily average and standard deviation
(std) was calculated from the differences of all 10 second measurements and the model daily output. For the comparison,
only the gird cell which would cover the respective buoy was considered. Since the resulting daily mean and std was still
too variable, it was further smoothened by a 7 days running mean. For the comparison of the fbRun, AWI and BGEP draft,
only weeks in which the AWI data covers the BGEP locations were considered. The model values are weekly means of the
respective buoy covering grid cell.

To be able to compare sea ice in situ measurements from more locations, the IMB buoy deployed during the MOSAiC
campaign are used (Lei et al., 2021). In contrast to the stationary measurements from the BGEP, the measurements drift along
the black trajectory in figure 1 from the center of the Arctic towards Greenland. The IMB buoy includes a thermistor string
reaching from the snow pack top to the ice-ocean interface at the bottom. A thermometer and a heating element are located
each 2 cm. The ice–snow, ice–water, and snow–air interface is measured, by heating the thermistor string up and measuring
the thermal response. More information on the instrument can be found in Jackson et al. (2013). The IMB buoys measure
the thickness of only one ice flow, unlike the BGEP upward looking sonar, and the data has a temporal frequency of one
measurement per day. To ensure that the comparison between the buoys and the gridded AWI sea ice thickness and model

output is reliable, 19 IMB buoys were considered. However, not all buoys were active at the same time. All buoys were
interpolated to the model grid by the nearest neighbor method.

For the comparison of the different model run vs. the IMB measurements, a minimum of 8 active buoys per day were chosen.
The limit of 8 buoys was chosen to account for the spatial coverage of the active buoys and at the same time secure a sufficient
number of days in which at least 8 buoys were active.

For the IMB sea ice thickness vs. assimilated sea ice thickness and the AWI sea ice thickness comparison, the IMB buoy
coverage of one week was projected on to the model grid, choosing the nearest neighbor. For the model data, only grid points
covered by the AWI data and the IMB buoys were chosen, and weekly averages were calculated for all three products. No
threshold of a minimum amount of active buoys was chosen, as this would have limited the available data too much.

## 3 Results

### 3.1 Freeboard and Sea Ice Concentration RMSE

To verify that the assimilation improves the modelled FB and SIC, the RMSEs between the assimilated data sets and the
model variables were computed after each assimilation time step. The calculation of RMSE includes all observed data points
of the assimilation time step. RMSE for FB is calculated on the available satellite tracks (marked orange in Figure 1), which
change every week, and the co-located model values. The same approach is used for SIC (the blue area in Figure 1) and the
corresponding model data.

The results are shown in the upper panels of Figure 3 and 4, and they are based on mean weekly model output data at the
location, where the corresponding observation exist. The lower panels in both figures show the difference between refRun
and sicRun respectively fbRun. Positive values indicate that the assimilation has improved the SIC or FB, and negative values
indicate that the variable was degraded by the assimilation. Degradation can occur when an assimilation variable disturbs the
physical balance of the model, and during a period of free run, when it is in the process of reestablishing its physical balance.

The results (Figure 3 upper panel) show that the reference run (black) had the highest RMSE of all and that the RMSE in-
creased the most over the assimilation period. This indicates that the assimilation improved the modelled sea ice concentration.
The RMSE for the assimilated runs (sicRun in turquoise and fbRun in orange) also increased over the assimilation period, but
to a lesser extent than the reference run. The lower panel in figure 3 shows a steady increase in the difference between the
reference run and the assimilated runs, reflecting the degree to which assimilation improved sea ice concentration.

The increase in RMSE over the season is a result of the chosen area for calculating RMSE and the definition of the metric
itself. RMSE weights larger errors more heavily than smaller errors. The FB differences are only calculated over areas with sea
ice, while the SIC data includes larger areas that are seasonally either ice-free or ice-covered. For SIC, the area with the largest
error, which is weighted most, is the ice edge at the Atlantic side, which increases over winter, accounting for the observed
seasonal increase in SIC RMSE from November to March in figure 3. Other assimilation studies have chosen to calculate
RMSE only over ice areas with sea ice concentration above 15% (Chen et al., 2017), but to be consistent, we chose to calculate
RMSE over the entire area.

The lower panel in figure 3 also shows negative values in October for the last two years, indicating that the assimilated runs agree less with the assimilated data than the reference run at the beginning of the assimilation period. The RMSE difference in the lower panel falls below 0 at the beginning of all assimilation periods after the initial one. As noted earlier, this can occur if the physical balance of the model is disturbed by assimilation. Figure 4 upper panel displays the RMSE of all FB values

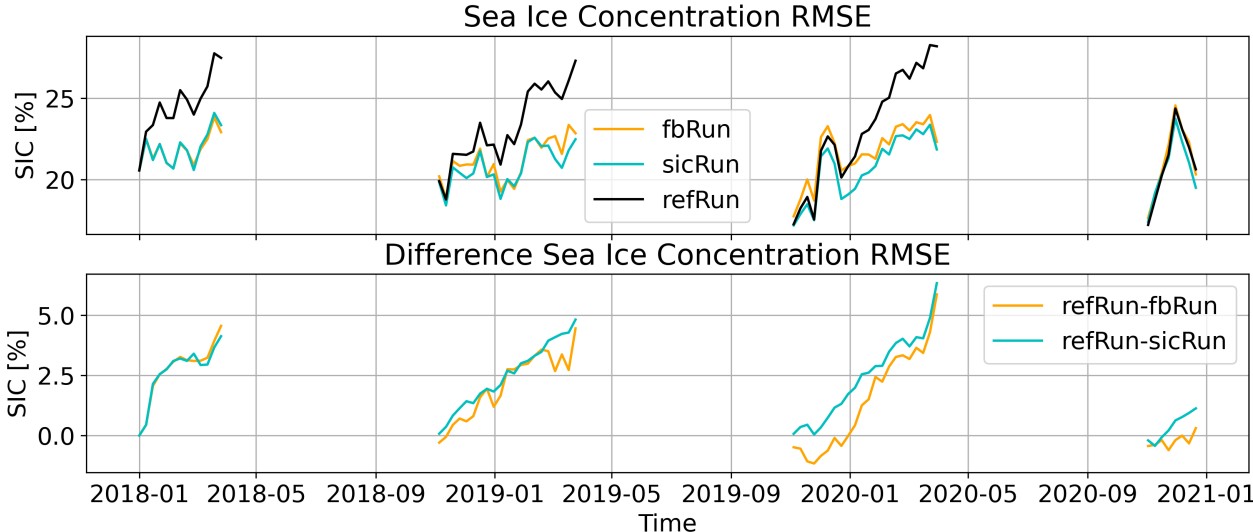

**Figure 3.** Top panel: Weekly SIC RMSE calculated at the observation data location, averaged over the corresponding assimilation time step. The orange graph shows the fbRun RMSE, the black the refRun and the turquoise the sicRun. Lower panel: The difference of the top panel RMSE of refRun-fbRun in orange and refRun-sicRun in turquoise.

assimilated at the corresponding time. The black line represents the refRun, while the turquoise line represents the sicRun. Both have almost equal FB RMSE throughout the assimilation period, ranging between 7 cm and 14 cm. The black refRun covers the turquoise sicRun in the upper panel. On the other hand, the FB RMSE for fbRun shows a clear drop within the first month of the assimilation period, reducing to about 5 to 6 cm. The lower panel in figure 4 shows that the RMSE differences

are all above 0, even at the beginning of a new assimilation period in November. It is expected that the SIC RMSE in figure 3 and the FB RMSE in figure 4 show improvements, as the observation values are used within the assimilation scheme, however this demonstrates that the assimilation works.

### 3.2 CryoSat-2 AWI sea ice thickness

To demonstrate that the sea ice thickness estimated through the FB assimilation method provides comparable results to other

sea ice thickness products derived from CryoSat-2, the sea ice thickness of fbRun was compared to the AWI sea ice thickness. The AWI sea ice thickness was selected because it is derived from the same FB values as the FB data assimilated in fbRun. Any differences between the two datasets therefore indicate the impact of the here introduced FB assimilation in contrast to the method of directly converting FB to sea ice thickness. Table 1 presents the correlation coefficients and biases for sea ice

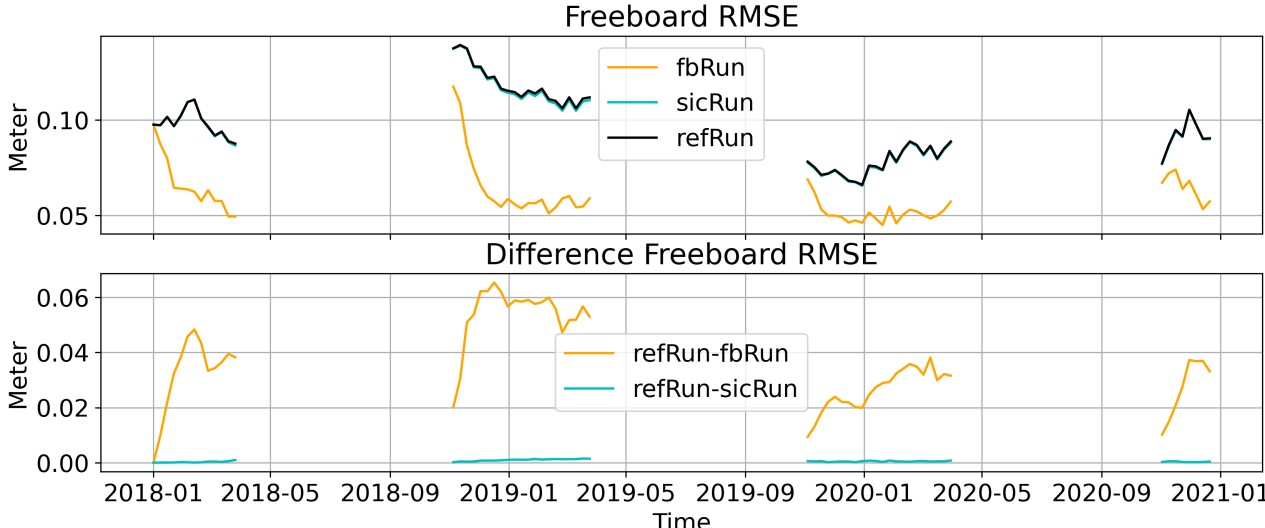

**Figure 4.** Top panel: Weekly FB RMSE calculated at the observation data location, averaged over the corresponding assimilation time step. The orange graph shows the fbRun RMSE, the black the refRun and the turquoise the sicRun. The black graph indicating refRun covers the turquoise graph indicating sicRun most of the time. Lower panel: The difference of the top panel RMSE of refRun-fbRun in orange and refRun-sicRun in turquoise.

**Table 1.** Monthly mean correlation coefficient and mean bias between the weekly AWI sea ice thickness (SIT) and FB and the fbRun SIT and FB for the entire assimilation period from 2018-01-01 to 2020-12-31. Only grid points covered by both the AWI FB data and the model were considered.

|  | October | November | December | January | February | March | April |
|---|---|---|---|---|---|---|---|
| Correlation coefficient SIT fbRun | 0.56 | 0.81 | 0.83 | 0.81 | 0.78 | 0.75 | 0.72 |
| Correlation coefficient SIT refRun | 0.40 | 0.49 | 0.45 | 0.44 | 0.44 | 0.51 | 0.50 |
| Bias SIT fbRun | -0.52 | -0.38 | -0.17 | -0.15 | -0.18 | -0.18 | -0.22 |
| Bias SIT refRun | -0.65 | -0.56 | -0.38 | -0.23 | -0.26 | -0.28 | -0.34 |
| Correlation coefficient FB fbRun | 0.30 | 0.68 | 0.79 | 0.76 | 0.78 | 0.78 | 0.74 |
| Correlation coefficient FB refRun | 0.06 | 0.09 | -0.2 | 0.2 | 0.05 | 0.16 | 0.19 |
| Bias FB fbRun | -0.03 | -0.02 | 0.01 | 0.01 | 0 | -0.01 | -0.02 |
| Bias FB refRun | -0.04 | -0.04 | -0.02 | -0.01 | -0.01 | -0.02 | -0.03 |

thickness and FB in refRun and fbRun compared to the AWI data. All spatially coinciding data points of the model runs and
the AWI data were considered over the entire period from 01-01-2018 to 31-12-2020. In general, the lowest correlations and
highest biases are found in October, as no data was assimilated yet and the assimilation period starts in November.

The sea ice thickness biases are negative for all months and runs, indicating that the modelled sea ice thickness and FB are thinner than the AWI data's FB and sea ice thickness. The sea ice thickness biases for both runs are smallest in January, and the FB biases are smallest in January and February. Overall, the FB biases are thinner than the SIT biases, which is no surprise as FB typically lies in the order of about 10% of sea ice thickness (Alexandrov et al., 2010).

Comparing the correlation coefficients of the refRun and fbRun for both the FB and sea ice thickness shows that the difference between the FB correlation coefficients is higher than the difference between the sea ice thickness correlation coefficients. This indicates that the FB assimilation brings the modelled FB closer to the assimilated FB data, but that the difference in deriving the SIT from the FB data also impacts the resulting SIT.

Figure 5 displays bivariate and univariate kernel density estimates (KDE) for sea ice thickness (panels a and b) and FB (panels c and d) for fbRun (in orange) and refRun (in blue) compared to the AWI data. The months of October and December were displayed as they represent the lowest and highest sea ice thickness correlation (see Table 1).

The KDE for both variables of fbRun changes from October to December, indicating higher correlation coefficients and smaller biases in December, which is a result of both thin and thick sea ice and FB getting thicker. However, the thicker FB and sea ice thickness values are still thinner than the AWI data variables, while the thin FB and sea ice thickness values are thicker than the AWI values. This could be a result of the assimilation discard negative FB values in the model, while the AWI data set includes negative FB values.

For the month following December (not displayed), the center of the sea ice thickness KDE, at about 1 m in figure 5 b), falls month by month further below the black regression line, while the thick sea ice thickness shows similar improvements compared to the refRun as the December plot. This indicates that the decreasing correlation and increasing bias (table 1) originate from the fbRun's sea ice thickness and FB becoming thinner compared to the AWI data sets values, while the thick sea ice compares equally well to the AWI sea ice thickness.

## 3.3 Upward looking sonar data

The BGEP upward looking sonar sea ice draft is independent of the satellite-derived FB data, and it is used for the comparison of the modelled sea ice draft, which is calculated as described in section 2.5. The BGEP data are not available for the complete period from 2018-01-01 to 2020-12-31, hence only data from October 2018 to December 2020 is used.

The BGEP ULS data, model data and AWI sea ice draft data are provided at different spatial and temporal coverage. To compare the different data sets, we split the comparison in two parts in order to account for these differences. In figure 6 the model draft from all three model runs are compared to the BGEP ULS drafts based on mean daily differences, whereas, figure 7 compares the AWI draft and the fbRuns draft with the BGEP ULS drafts based on mean weekly differences only at locations covered by the AWI data.

The differences between the BGEP upward looking sonar ice draft and the model sea ice draft are shown in figure 6. The dashed line shows the fbRun, the solid line the refRun, and the dotted dashed line the sicRun. The gray shaded areas indicate the assimilation period. For all three moorings, fbRun shows the values in closest agreement with the observations throughout the entire period displayed. This is also reflected by the lower RMSE listed in table 2 The refRun and the sicRun are almost

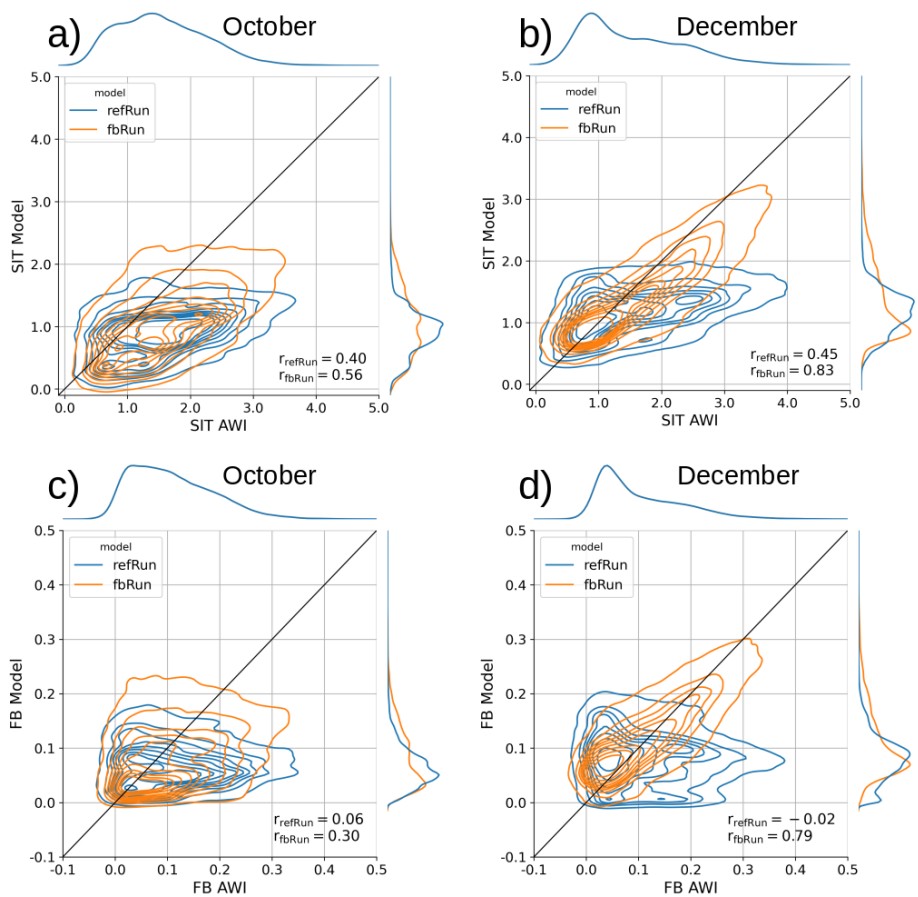

**Figure 5.** The bivariate and univariate kernel density estimate (KDE) for sea ice thickness and FB for the model runs fbRun and refRun in comparison to the AWI sea ice thickness and FB. a) and b) show the sea ice thickness in October and December and c) and d) show the FB for October and December. The month October and December were chosen because October is the month with the lowest sea ice thickness correlation between fbRun and AWI (as listed in table 1). The correlation coefficients r are displayed in the lower right corner of each plot. The black line indicates r=1 and the units are in meter.

in perfect agreement except for a few days as for example in October 2019 at BGEP mooring A and D. The RMSE between the BGEP data and the fbRun is with 0.41 m 23 cm lower than the RMSE of refRun and sicRun. Periods in summer, when the observation std is 0 m, indicates periods with no ice present in the observations. Gaps indicate periods where no data are available. The BGEP observations are all ice free in summer 2019 while only fbRun at BGEP mooring A reaches the point of being ice free in late September until beginning of November 2019.

Figure 7 shows the mean differences between the AWI sea ice draft and the fbRun sea ice draft. To do so, the AWI data set was interpolated to the model grid and only data points covered by all three data sets (AWI CryoSat-2, fbRun and BGEP) were considered. Instead of daily averages as shown in figure 6, weekly averages were calculated, since the AWI sea ice draft

**Table 2.** Mean bias and RMSE calculated between the BGEP ULS draft measurement and the model runs fbRun, sicRun and refRun and the MOSAiC IMB sea ice thickness and the model runs. The RMSE and biases were calculated for all three mooring locations together, the assimilation period marked gray in figure 6 and the free run period.

|  | BGEP ULS total | MOSAiC IMB |
|---|---|---|
| RMSE fbRun | 0.41 m | 0.20 m |
| RMSE sicRun | 0.64 m | 0.09 m |
| RMSE refRun | 0.64 m | 0.10 m |

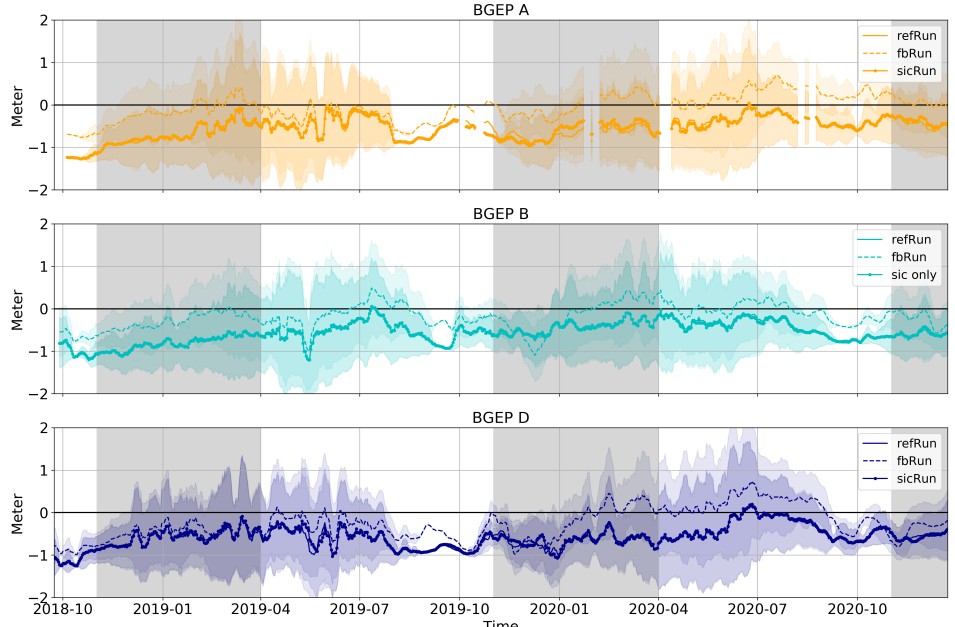

**Figure 6.** Daily mean sea ice draft differences and std between BGEP observation and all three model runs. The shaded colored area shows one std calculated for each day from the 10 seconds record. The std appears darker where the std from different model runs overlap. The gray shaded area indicates the assimilation period. The dashed line shows observed sea ice draft minus fbRun sea ice draft, the solid line the refRun and the dot-dashed line the sicRun only. The upper panel shows data from mooring A, the middle panel data from mooring B and the lower one from mooring D. The sites are marked in the corresponding colors in fig 1.

is provided in weekly time steps. The dashed lines in figure 7 show the AWI data, and the solid lines the fbRun data. The gray background shows the assimilation period. Colors are chosen per mooring according to figure 1. The resulting differences

**Table 3.** The mean RMSE of the weekly mean differences shown in figure 7. The RMSE was calculated on average for each mooring and both the fbRun ice draft and the AWI CryoSat-2 ice draft. All values are given in meters.

|  | BGEP mooring A, B, D | MOSAiC IMB |
|---|---|---|
| fbRun | 0.30 m | 0.23 m |
| AWI CryoSat-2 | 0.30 m | 0.34 m |

between the fbRun and the AWI CryoSat-2 sea ice draft are shown in figure 7. Both the AWI sea ice draft and the fbRun sea ice draft differ about $\pm$ 50-90 cm from the mooring data. There is no clear bias, or seasonality in either differences, and they don't always follow the same pattern, except in winter 2019/2020 where both data sets begin with a negative bias and end with a positive bias, with the exception of a few weeks in the AWI CryoSat-2 draft in the end of the assimilation period.

The RMSEs between the BGEP moorings sea ice draft, the fbRun sea ice draft and AWI CryoSat-2 sea ice draft were calculated. They are listed in table 3 The RMSEs of the data products compared to the mooring data are both 0.3 m.

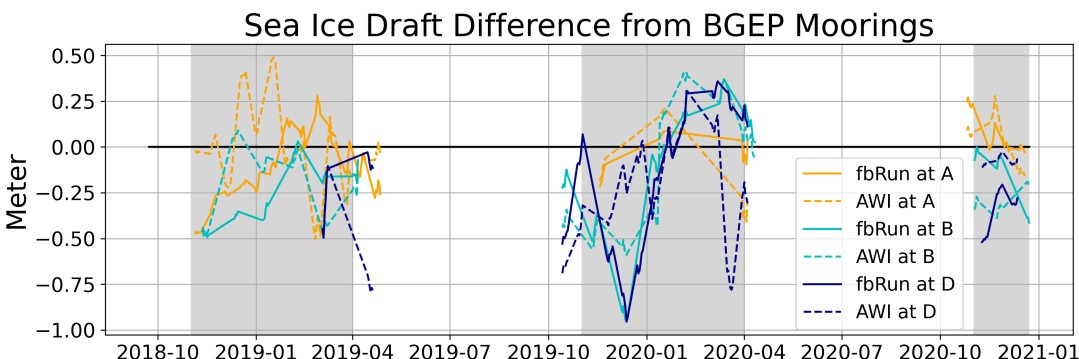

**Figure 7.** The weekly mean difference between the BGEP upward looking sonar sea ice draft measurements and sea ice draft calculated from the AWI sea ice data set (dashed lines) and the fbRun sea ice data (solid lines). The color indicate the location in figure 1. Positive values indicate that the BGEP draft is thicker.

### 3.4   MOSAiC IMB data

The MOSAiC data covers a different spatial area than the BGEP observations. Data are interpolated to daily and weekly means respectively in order to have the same frequency as the data that they are being compared to. Details are described in section 395   2.5.

In figure 8, the daily sea ice thickness from the MOSAiC IMBs and the three model runs are plotted for days when at least 8 buoys were active. The shaded area around each line indicates one std of the respective displayed data. The MOSAiC IMB dataset has the largest std and all model runs lies within this std for most of the observation period, with an exception to the

fbRun's sea ice thickness in October 2019, April 2020 and June 2020. Overall, the modelled, assimilated, and observed sea ice thickness grow over the same period from October 2019 to April 2020, and all four sea ice thickness also start to decline at about the same time in June 2020. The observed sea ice thickness starts to be more variable in the beginning of June 2020, which is not reflected in the model data. The variability in the observation data is most likely caused by the reduced number of buoys being active during this time and the sea ice being more mobile as it starts to melt. Both the refRuns and sicRuns sea ice thickness compare better to the MOSAiC observation than the fbRun. This is also reflected in the RMSE calculated for the fbRun, sicRun, and refRun in comparison to the MOSAiC sea ice thickness in Table 3. A one-sided t-test was performed, comparing the differences between the different model runs and the MOSAiC IMB sea ice thickness. The one-sided t-test showed that the sicRun's and refRun's sea ice thickness RMSE was significantly lower than the fbRun's RMSE.

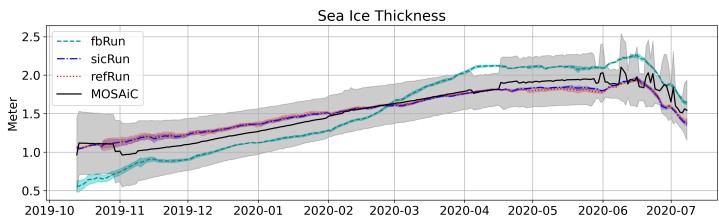

**Figure 8.** Daily mean sea ice thickness averaged over all grid cells covered by at least 8 buoys active per day. The black solid line indicated the MOSAiC IMB measured sea ice thickness, the red dotted line indicates the refRun sea ice thickness, the blue dashed dotted line the sicRun sea ice thickness and the turquoise dashed line the fbRun sea ice thickness. The shaded areas around each of the graphs indicate one std of each daily averaged sea ice thickness data set.

Figure 9 shows the weekly mean sea ice thickness from the MOSAiC IMBs and the three model runs. The average is calculated as described in section 2.5. The yellow dashed-dotted line represents the AWI sea ice thickness, the turquoise dashed line represents the fbRuns sea ice thickness, and the black solid line represents the MOSAiC sea ice thickness. The transparent shaded background in each corresponding color indicates one std. All three sea ice thicknesses increase over the displayed period. The AWI sea ice thickness increases the most from approximately 0.6 m to 2.3 m with a sharp drop in the last week of April. The MOSAiC data displays less growth and start slightly thicker than both the fbRun and AWI sea ice thickness at around 0.8 m in October 2019 and reaches around 1.8 m in April 2020.

When comparing the sea ice thickness for the fbRun from figures 8 and 9 a), it is apparent that the fbRuns sea ice thickness follows a similar pattern. However, this is not the case for the MOSAiC sea ice thickness. Comparing the sea ice thickness for the MOSAiC IMB data from figures 8 and 9 a), the data in 9 a) appears to be more. This difference is caused by the amount of buoys considered. The buoys considered in figure 9 depend on the sparse AWI data coverage, while 8 considers at least 8 buoys per day. This leads to larger jumps from week to week of the MOSAiC sea ice thickness in figure 9 than in figure 8. This is also evident by the low std at the beginning of March and mid-April 2020 in figure 9.

Table 3 lists the RMSE calculated between the AWI and the MOSAiC sea ice thickness and between fbRun and MOSAiC sea ice thickness. The RMSE calculated for the AWI sea ice thickness is 11 cm greater than the RMSE calculated for the fbRun

sea ice thickness. A one-sided t-test was performed to determine the statistical significance of the difference, which showed that the fbRuns RMSE is significantly smaller than the AWI RMSE.

In figure 9 b), the radar FB for the refRun, fbRun, and AWI data are shown. The fbRun and AWI data FB in figure 9 a) and the respective sea ice thickness in 9 b) do not entirely follow the same pattern. The AWI FB starts out thinner than the fbRun's FB, while the AWI sea ice thickness is thicker than the fbRun's sea ice thickness throughout the entire displayed period. This indicates that the difference is caused by the difference in snow thickness and sea ice density. The AWI data are the FB values that were assimilated, and the fbRun FB is approximately between the refRun's and AWI data values, showing the effect of

the assimilation. It is clear from figure 8 that the refRun and the sicRun are closer to MOSAIC IMB data, however figure 9 shows that the fbRun follows the evolution of the observed radar FB better. This shows that the assimilation act as expected, but in this area there is a discrepancy between the in situ observations from MOSAIC IMB and the remotely sensed AWI FB observations. The relation between the FB from refRun and fbRun follows a similar pattern as the sea ice thickness in figure 8, since the sea ice density, snowfall, and water density values are not significantly influenced by the assimilation.

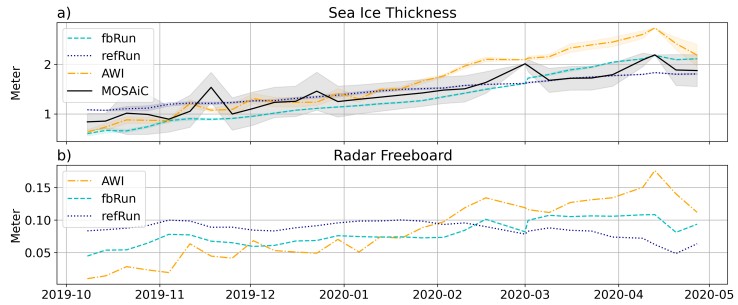

**Figure 9.** a) Weekly mean sea ice thickness averaged over all grid cells covered by the CryoSat-2 flight pass considered in the AWI data set. The mean sea ice thickness is displayed with one std for sea ice thickness from MOSAiC (black solid), AWI (yellow dashed dotted), fbRun (turquoise dashed) and refRun (dark blue dotted). b) as a) but for radar freeboard and without MOSAiC observations. The yellow dashed dotted line shows AWI, the turquoise dashed line the fbRun and the dark blue dotted line refRun radar FB.

## 4   Discussion

To show the effect of the assimilation, the RMSE between the assimilated SIC and FB observations and the modelled SIC and FB was calculated for refRun, sicRun and fbRun. Figure 3 and 4 show that SIC and FB as expected are improved in each winter season when satellite derived FB and SIC are assimilated. Further, the correlation coefficient between the AWI FB data (which was assimilated) and the fbRun FB data is higher than the correlation coefficient of the refRun and the AWI FB data.

The fbRun's sea ice thickness correlations and biases in table 1 also indicates a closer agreement with the AWI data when compared to the refRun's correlations and biases. This show that the FB assimilation has an effect on the modelled sea ice thickness.

The RMSE between the assimilated SIC and FB observations and the modelled SIC and FB was calculated for refRun, sicRun, and fbRun, as shown in figures 3 and 4. The results show that assimilation of satellite-derived sea ice concentration and freeboard data has a positive effect on the model performance, with improved sea ice concentration and freeboard values in each winter season. The fbRun's sea ice thickness, FB correlations and biases in Table 1 suggest closer agreement with the AWI data than the refRun's correlations and biases. This again shows that the FB assimilation has an effect on the modeled sea ice thickness.

The comparisons to independent sea ice thickness observations indicate that the fbRun sea ice thickness is improved in the Beaufort Sea, but not in the central Arctic. In contrast, refRun and sicRun perform significantly better in the central Arctic. Notably, the in situ observations in the Beaufort Sea cover more than two years, while those in the central Arctic only cover nine months. The RMSE plots in figure 4 show that refRun's RMSE during the winter of 2019/2020 is lower than in the prior month. Moreover, the calculation of the mean sea ice thickness difference between the refRun and the fbRun at the location of the MOSAiC IMB data in October for other years showed that 2019 was the year with the largest differences. This indicates that the sea ice thickness in this region is highly variable and suggests that the better performance of the refRun and sicRun in winter 2019/2020 might not be representative for all years. The FB values in figure 9(b) could suggest that the assimilated FB data causes the thinner ice for the fbRun sea ice thickness in figure 8. The assimilation begins in November, when the fbRun's sea ice thickness is already thinner than the refRun's and sicRun's sea ice thickness. Thus, the thinner sea ice in figure 8 is a result of the assimilation in the previous year. To be able to compare the year 2019 with other years, the mean sea ice thickness differences between the refRun and fbRun were calculated at the location of the MOSAiC IMB data in October. The mean difference between the refRun and the fbRun is 28 cm for 10-2018, 50 cm for 10-2019, and 2 cm for 10-2020. The MOSAiC year is clearly the one with the largest difference.

Considering the refRun's RMSE in other years, the inter annual variability of sea ice thickness in the examined region, the fact that the observations in the Beaufort Sea span a significantly longer time, and the fact that the BGEP ULS fbRun RMSE is over 20 cm lower than the refRun RMSE and only 10 cm higher for the MOSAiC IMB locations, we argue that the fbRun's sea ice thickness is overall improved in comparison to the sicRun's and refRun's sea ice thickness. Nevertheless, the difference between the Beaufort Sea and the central Arctic in the observations and the model runs underlines the need for more long-term in situ observations.

Dirkson et al. (2017) and Day et al. (2014) show that SIC has a shorter memory than sea ice thickness. The facts that, FB improves sea ice thickness, as shown in figure 6, and that FB values are still improved after summer in all years (in contrast to SIC) as shown in figure 4 lower panel suggests that FB also keeps the memory as opposed to SIC.

The AWI sea ice thickness could be a typical CryoSat-2 product that could be assimilated in order to improve the modelled sea ice thickness. Based on the RMSE in table 2, which show that the FB assimilation give better values compared to the MOSAiC data and similar results in the Beaufort Sea, the method presented in this study show the perspective of assimilating FB instead.

We discussed that the thinner fbRun sea ice thickness in October in figure 9 and 8 is not caused by assimilating the also thinner AWI FB, as the assimilation starts in November. In contrast, the significantly larger increase in fbRun's sea ice thickness

later in the year is a direct result of assimilating thick FB: In the second half of the 2019/2020 winter season, the AWI sea ice thickness (figure 9 a)) was clearly thicker than the MOSAiC sea ice thickness. While it is not as clear for the fbRun's sea ice thickness in figure 9 a), figure 8 clearly shows that the fbRun's sea ice thickness is also thicker than the MOSAiC sea ice thickness. The increase in fbRun's sea ice thickness during late February to early April 2020 (figure 8) follows the increase in AWI FB (yellow line in figure 9 b) starting in end January 2020. Since the AWI FB is assimilated in fbRun, this increase is caused by the assimilation. However, this assimilation leads to sea ice that is too thick, as seen in figure 8. This overestimation of sea ice thickness is likely due to an overestimation of FB in the AWI data, as found by King et al. (2018) in their field campaign in April. Other studies (Giles and Hvidegaard (2006), Willatt et al. (2011), and Ricker et al. (2015)) suggest similar biases in the radar backscattering horizon for deep snow and high moisture content. Giles and Hvidegaard (2006) and King et al. (2018) both conducted field studies in March and April, months when the assimilated AWI FB (figure 7 b)) is highest, near the final MOSAiC location. The resulting overestimation of sea ice thickness in the AWI data and the comparable, thinner assimilated sea ice thickness from fbRun is a good example of the advantage of assimilating FB instead of sea ice thickness.

The increase in biases and the decrease in correlations shown in Table 1 exhibit a similar pattern as the FB and sea ice thickness at the MOSAiC IMB locations discussed above. This similar behavior could indicate that the pattern displayed in figure 7 is not restricted to the observation area, and suggests that the FB assimilation could correct the error introduced by the wrongly located scattering horizon in the CryoSat-2 FB retrievals to some extent. However, the thickness comparison of fbRun and AWI data to the BGEP data set (figure 6 and figure 7) does not show the same seasonal pattern in thickness discussed above for the MOSAiC observation. This might indicate regional differences in the scattering horizon or that the assimilation does not correct for the effect everywhere in the same manner. Further studies are needed to investigate this.

## 5   Conclusions

In this study, a method to assimilate FB is described, and the results from a 3 years assimilation run is evaluated. The presented method builds upon calculating an increment using modelled FB and then converting the changed FB into the sea ice thickness. The method uses parameters from the sea ice model for the sea ice density, snow density and snow thickness instead of the prescribed values used in the AWI sea ice thickness product, which it is compared to. First, it was shown that the FB assimilation improves the modelled FB (figure 4) and that the assimilation effects the sea ice thickness (table 1). Figure 6 shows that the sea ice thickness of the run assimilating FB is improved in the Beaufort sea. The comparison to MOSAiC IMB sea ice thickness data from the central Arctic does not give the same results. Here the refRun and sicRun perform better, but we can show that the poorer performance of the assimilation is to some extent due to too thick FB being assimilated. CryoSat-2 FB is known to have a thick bias in late winter due to uncertainties in the back scattering horizon of the radar signal (Giles and Hvidegaard, 2006; Willatt et al., 2011; Ricker et al., 2015). The seasonality of the biases and correlation listed in table 1 as well as the observation comparison in figure 9 indicate that the assimilation has some skill in mitigating this bias. One of the two main objectives was to determine if the FB assimilation improves sea ice thickness. Even though fbRun compares worse to the MOSAiC IMB observations than the refRun, fbRun is in closer agreement with the longer observation record at the BGEP locations.

To compare our method to sea ice thickness data from a more classical approach, we have chosen the weekly sea ice thickness product from the AWI sea ice portal (Hendricks et al., 2021). This sea ice thickness is derived from the same FB as assimilated in fbRun. Overall, the AWI CryoSat-2 sea ice thickness and FB is thicker than the fbRun's sea ice thickness and FB (table 1). When comparing the two sea ice thicknesses to an independent sea ice measurements from the BGEP upward looking sonar data, we can show that the FB assimilated sea ice thickness and AWI sea ice thickness result in similar RMSEs. The comparison to sea ice thickness observations from MOSAiC IMBs deployed during the MOSAiC in the central Arctic result in significantly lower RMSE for the sea ice thickness from the FB assimilation.

## 5.1 Outlook

The presented method builds upon modelling the most influential variables of equation 6. These are the snow thickness, the snow density and the sea ice density (Alexandrov et al., 2010). The snow density used in this study does not differ from the snow density used in the AWI data product. The results in figure 7 show that the modelled variables result in comparable results at the BGEP locations and better results in the central Arctic as the empirical values used in the AWI sea ice thickness product. Both the snow thickness and the sea ice density differ, and no clear conclusion can be drawn at this point, whether the AWI values or the model values are more correct. As the aim of this study was to present the method on how to assimilate FB and a validation of the resulting sea ice thickness, a detailed discussion of the model parameter and the resulting influence on the sea ice thickness when compared to more traditional approaches is not included. A study with a focus on this is currently in preparation.

*Code availability.* The CICE code is available from git. The NEMO code is available from here. The PDAF code can be downloaded from this homepage. Additional CICE routines for the FB assimilation are available upon request, from the contact author.

*Author contributions.* IS conceived the assimilation set up, implemented it and wrote the manuscript draft. TAR edited and reviewed the manuscript and advised on matter related to the assimilation set up and CICE. LS edited and reviewed the manuscript and advised on CryoSat-2 related matters.

*Competing interests.* To the knowledge of the authors there are no competing interest

*Acknowledgements.* The data were collected and made available by the Beaufort Gyre Exploration Program based at the Woods Hole Oceanographic Institution (https://www2.whoi.edu/site/beaufortgyre/) in collaboration with researchers from Fisheries and Oceans Canada at the Institute of Ocean Sciences.

This study is a collaboration between the Danish Meteorological Institute, Aalborg University and the Danish Technical University. It is funded by the Danish State through the National centre for Climate Research and the Act of Innovation foundation in Denmark through the MARIOT project (Grant Number 9090 00007B)

The model input contains Copernicus Climate Change Service information (2021) and neither the European Commission nor ECMWF is responsible for any use that may be made of the Copernicus information or data it contains.

The OSI SAF Sea Ice Index v2.1 is made available at https://osisaf-hl.met.no/v2p1-sea-ice-index

We thank Lars Nerger for his help during the implementation process of PDAF.

We would also like to thank two anonymous reviewers for their comments, helping us improve the presented study significantly.

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
