# Peer review of "Assimilating CryoSat-2 freeboard to improve Arctic sea ice thickness estimates"

_The Cryosphere, 2022_

## Referee Comment (RC1)

**Summary and Decision:**

The manuscript "Assimilating CryoSat-2 freeboard to improve Arctic sea ice thickness estimates" by Sievers et al. presents a new study in which satellite-derived radar freeboard (FB) from the Alfred Wegener Institute (AWI), and sea ice concentration (SIC) from the Ocean and Sea Ice Satellite Application Facility (OSI-SAF) are assimilated into the CICE sea ice model in the Arctic, between the period 2018-2020. To benchmark the improvements gained from assimilating FB, comparisons are made to an experiment which assimilates only SIC, and another experiment in which no assimilation is performed. RMSE validation across the three experiments show that modelled FB is improved by assimilating FB and SIC observations, while no improvement in FB is obtained by only assimilating SIC. Comparing observations of sea ice thickness to thickness from the FB assimilation experiment shows that the representation of thicker ice is improved for a test case in March 2020. On the other hand, a snapshot example over the same period suggests that sea ice thickness after FB assimilation is now too low in the Canada basin. Comparisons of sea ice draft are also made with 3 separate moorings from the Beaufort Gyre Exploration Project (BGEP), where sea ice draft in the FB assimilation experiment is consistently improved over the 2018-2020 period, relative to the SIC only assimilation and reference experiments.

The notion of assimilating radar FB, as opposed to sea ice thickness, is well motivated, given the large uncertainties involved when converting FB to thickness, and the authors provide a good overview of this topic in the opening sections. I do have some concerns however relating to the clarity of the methods and the rigor of the validation, which I feel need to be addressed. The methods section in particular is difficult to follow, and the lack of details on the model experiments mean that reproducibility is an issue. Relating to the validation, at present it is difficult to say how well the assimilation is performing in a) different regions of the Arctic, and b) different times of the year. For example, is the thin ice in the Canada basin after FB assimilation a systematic feature throughout the 2018-2020 period? Or does this just occur in the one snapshot? I like the comparison to BGEP moorings as this shows a clear win for the FB assimilation at these locations. It would also be useful however to see e.g., monthly-mean spatial RMSE plots and time series comparisons (see some of my suggestions below). On this note, I'm also unsure why the authors have limited themselves to such a short period (2018-2020), when both CryoSat-2 and BGEP data are available back to 2010. I would strongly encourage the authors to extend their study to this full period in order to give more confidence that modelled thickness is indeed improved by assimilation of FB. I realise that this would create significantly more work and so may be an unrealistic request. Perhaps if some additional analysis shows convincingly that the assimilation is doing a good job between 2018-2020, then extending to 2010 will not be necessary.

In any case, I feel there is a fair bit of work needed before I can recommend this manuscript for publication. Therefore, I recommend major revisions for this article. My thanks to the authors for their work and I look forward to reading the next version!

**General Comments:**

**Introduction**

- The authors have done a good job at summarising the various uncertainties/assumptions related to deriving sea ice thickness estimates (L26-86), however one key piece of missing information is the choice of retracking algorithm. The roughness characteristics of the sea ice cause different degrees of scattering of the radar echo, which are then convoled to produce an average height of the snow-ice interface. A retracking algorithm which does not account for changes in scattering due to roughness may therefore produce a freeboard which is too high when sea ice roughness is high, and vice versa. Landy et al., 2019 for example have shown how the use of a 'physically-based' retracker can help mitigate these effects. I think the introduction section here should include a few sentences to highlight this as a source of uncertainty in sea ice thickness estimates.
- L81: I'm a bit wary of saying that by assimilating FB, the effects of snow thickness and density errors are eliminated. Sea ice radar FB assumes that the radar echo is returned from the snow-ice interface, and this generally is not the case (e.g., Willatt et al., 2011; Nab et al., 2023). To appropriately model the scattering surface of the radar echo (and hence reduce uncertainty in FB) we need to account for snow thickness, density and other dieletric properties of the snow. Maybe just worth highlighting this as another source of uncertainty in satellite-derived sea ice thickness.
- Figure 1 (and others throughout the manuscript): I suggest changing colours from red and green (in this case, the BGEP locations) to something more colour-blind friendly.

**Methods and data**

- I find section 2.2 a little hard to follow and am also struggling to relate it to section 2.6. Is it essential to have these as separate sections? Can section 2.2 not be merged in with section 2.6? In any case, it would be useful to provide more details about the various model runs and how they were initialised etc, and potentially updating figure 2 with more information. For example, what are 'VAR' and 'VARI'? I will summarise what I think I understand, and please correct me if I'm wrong:
  - An initial experiment was run between 1995-2020. This experiment was run as an 80-member ensemble in coupled ice-ocean mode, and forced by ERA5 atmospheric reanalysis
    - What were the initial ice/ocean conditions for this run?
    - How do you e.g., perturb the ice/ocean model parameters to create the ensemble?
    - I'm not sure what is meant by increasing the variance to "account for biases" (L138). Are you not just increasing the variance to prevent ensemble collapse? Ultimately, you're hoping that the data assimilation itself will reduce the biases
  - The 2018-2020 period of the initial experiment corresponds to the refRun

- o The initial experiment at 2018-01-01 was used as initial conditions for both the sicRun and the fbRun. Assimilation over the 2018-2020 period is performed every 7 days.
  - o The increments from the assimilation runs are then saved, and then you effectively re-run the sicRun and fbRun experiments over the 2018-2020 period, except that the previously saved increments are now updating the model at every time step (through linear interp of the increments from 7 days to 600 sec).
    - ▪ This is to prevent model shock after each assimilation cycle?
  - o I'm not sure where the 'static ensemble' fits into all of this? Could you explain?
- More generally, could you explain the motivation for only focusing on the 2018-2020 period rather than the entire CryoSat-2 period (2010-present)? By utilising the entire record I feel that you would be able to derive more rigorous statistics related to the improved FB and thickness from assimilation. For example, time series comparisons of monthly-mean FB and thickness between AWI and the assimilation run over the 2010-present period, for different Arctic regions.
- L135: Could you provide more details on why you choose observational error estimates of 15% for SIC and 0.15m for FB, given that on L175 and L166 you state that the observational uncertainties are 10% and <0.07m, respectively?
- L150-155: Is there any post-processing applied after assimilation to ensure that the updated SIC is bounded between 0 and 1? If so, how is this bounding applied? Particularly to the category terms.
- I believe currently in the fbRun you are updating SIC and FB sequentially. Out of curiosity, is sea ice thickness updated during the assimilation of SIC, and similarly is SIC updated in the assimilation of FB? Do you also expect your results to differ if you first assimilate FB and then SIC?

**Results**
- L269-270: Suggest clarifying here that by "assimilation period" you mean November-March, as opposed to the whole 2018-2020 period.
- L287: Does "beginning of October" and "end of winter" refer to a single day? Or a weekly average? Please clarify.
- Figure 5: Suggest including either correlation or $R^2$ values for both refRun and fbRun on each plot to quantify the improvement. I also suggest making this a 4-panel figure, and show the equivalent density plots for freeboard as well.
- L288: Can you also speculate why the "thin bias" problem is not improved with the assimilation of FB? Is this because the FB increment is spread linearly across the ITD categories (Equation 1), whereas in reality more weight should be given to the thinner categories?
- Figure 6: I don't think panels C) and D) are all that informative. Especially considering that our observational estimates of snow thickness and ice density are inherently wrong. I suggest replacing these panels with freeboard comparisons (AWI-refRun) and (AWI-fbRun). It would also be good on each panel to include a mean RMSE value, as it currently looks as if panels A) and B) might actually be similar in terms of RMSE.

- Figure 8: Could you speculate why the differences between AWI and fbRun are significantly large at the beginning of the 2018-2019 period? Is this a spin-up issue?

**Discussion**
- L400-410: Is the imprint of the FYI/MYI zone on the difference plot not just due to the fact that the AWI snow thickness and ice densities are assigned constant values depending on whether FYI or MYI? In your case you've replaced the density with a linear function after Mallett et al., 2020, but ultimately this imprint of a FYI/MYI mask is just a reflection of what the observations are (hence my suggestion above to remove).
- L412: The panels in Figure 6 are errors in modelled sea ice thickness relative to AWI, not uncertainties in AWI sea ice thickness, right?

**Minor suggestions:**

**Abstract**
L4: Define SIC and FB acronyms
L6: Define AWI acronym
L10-11: This last sentence is rather vague. I suggest something like "Modelled sea ice draft errors are in good accordance with that of CryoSat-2 errors at BGEP mooring locations, with mean error differences less than 3 cm over the 2018-2020 period."

**Introduction**
L17: I'm not sure what is meant here by "need to affect the model variable that the assimilation aims to improve". Does it mean that we need to use an appropriate observational operator to map the model state variable which is being updated, to the space of the observations? E.g., an operator to map thickness to freeboard?
L18: I would clarify here that this statement relates specifically to Arctic sea ice predictability on seasonal-to-interannual timescales.
L24: Suggest stating explicitly that initialising thickness is better for predicting Arctic sea ice on seasonal time scales or longer (sea ice area persistence is more important at short lead times).
L55-57: Would also make reference here to the recently developed SnowModel-LG (Liston et al., 2020; Stroeve et al., 2020), which is being adopted in sea ice thickness products from e.g., Landy et al., 2022.
L65: The phrasing "add up" here feels a little vague. Does this mean that the error in sea ice thickness is equal to a linear sum of the errors in snow/ice/water density and FB?
L71: Suggest rephrasing to "The OSI-SAF ice type product (Aaboe et al., 2021) is one observational data set which aims to distinguish between FYI, MYI and ambiguous ice types."
L76-77: The wording is a bit confusing here. Does it mean that the errors are systematically overestimated in the MYI zone and underestimated in the FYI zone in the OSI-SAF product?
L77-78: As far as I'm aware, sea ice area isn't required to generate thickness anyway, so this sentence seems redundant. Could you explain what you mean here? Unless you're referring to CryoSat-2-derived sea ice volume?

**Methods and data**

L96: First time using PDAF acronym, please define.

L99: Suggest rephrasing to "An increment is the amount of change in a model state variable after the assimilation of observational data."

L106: First time using WMO acronym, please define.

L109: Suggest removing first sentence on L109 and changing L111 to "the key variables are snow thickness ($h_s$), snow density ($rho_s$), sea ice density ($rho_i$), and ocean water density ($rho_w$)." Or similarly just defining the terms explicitly on L109.

L142: 80 model states? Or 80 ensemble members?

**Results**

L299: Suggest clarifying that the date 03-30-2020 is actually a 7-day mean.

L315: change to "independent of the satellite-derived FB data".

L330: change here and elsewhere in manuscript from "data is" to "data are"

**Discussion**

L410: I'm not sure on Crysophere reference guidelines, but you might need to change "Sievers et. al (in preparation)" to something like "Future work will analyse the effects of different variables…"

**References**

Landy, J.C., Tsamados, M. and Scharien, R.K., 2019. A facet-based numerical model for simulating SAR altimeter echoes from heterogeneous sea ice surfaces. *IEEE Transactions on Geoscience and Remote Sensing*, *57*(7), pp.4164-4180.

Willatt, R., Laxon, S., Giles, K., Cullen, R., Haas, C. and Helm, V., 2011. Ku-band radar penetration into snow cover on Arctic sea ice using airborne data. *Annals of Glaciology*, *52*(57), pp.197-205.

Nab, C., Mallett, R., Gregory, W., Landy, J., Lawrence, I., Willatt, R., Stroeve, J. and Tsamados, M., 2023. Synoptic variability in satellite altimeter-derived radar freeboard of Arctic sea ice. *Geophysical Research Letters*, p.e2022GL100696

Liston, G.E., Itkin, P., Stroeve, J., Tschudi, M., Stewart, J.S., Pedersen, S.H., Reinking, A.K. and Elder, K., 2020. A Lagrangian snow-evolution system for sea-ice applications (SnowModel-LG): Part I—Model description. *Journal of Geophysical Research: Oceans*, *125*(10), p.e2019JC015913.

Stroeve, J., Liston, G.E., Buzzard, S., Zhou, L., Mallett, R., Barrett, A., Tschudi, M., Tsamados, M., Itkin, P. and Stewart, J.S., 2020. A Lagrangian snow evolution system for sea ice applications (SnowModel-LG): Part II—Analyses. *Journal of Geophysical Research: Oceans*, *125*(10), p.e2019JC015900.

Landy, J.C., Dawson, G.J., Tsamados, M., Bushuk, M., Stroeve, J.C., Howell, S.E., Krumpen, T., Babb, D.G., Komarov, A.S., Heorton, H.D. and Belter, H.J., 2022. A year-round satellite sea-ice thickness record from CryoSat-2. *Nature*, *609*(7927), pp.517-522.

---

## Author Response (AR1)

Reviewer 1:

Summary and Decision:

The manuscript "Assimilating CryoSat-2 freeboard to improve Arctic sea ice thickness estimates" by Sievers et al. presents a new study in which satellite-derived radar freeboard (FB) from the Alfred Wegener Institute (AWI), and sea ice concentration (SIC) from the Ocean and Sea Ice Satellite Application Facility (OSI-SAF) are assimilated into the CICE sea ice model in the Arctic, between the period 2018-2020. To benchmark the improvements gained from assimilating FB, comparisons are made to an experiment which assimilates only SIC, and another experiment in which no assimilation is performed. RMSE validation across the three experiments show that modelled FB is improved by assimilating FB and SIC observations, while no improvement in FB is obtained by only assimilating SIC. Comparing observations of sea ice thickness to thickness from the FB assimilation experiment shows that the representation of thicker ice is improved for a test case in March 2020. On the other hand, a snapshot example over the same period suggests that sea ice thickness after FB assimilation is now too low in the Canada basin. Comparisons of sea ice draft are also made with 3 separate moorings from the Beaufort Gyre Exploration Project (BGEP), where sea ice draft in the FB assimilation experiment is consistently improved over the 2018-2020 period, relative to the SIC only assimilation and reference experiments.

The notion of assimilating radar FB, as opposed to sea ice thickness, is well motivated, given the large uncertainties involved when converting FB to thickness, and the authors provide a good overview of this topic in the opening sections. I do have some concerns however relating to the clarity of the methods and the rigor of the validation, which I feel need to be addressed. The methods section in particular is difficult to follow, and the lack of details on the model experiments mean that reproducibility is an issue. Relating to the validation, at present it is difficult to say how well the assimilation is performing in a) different regions of the Arctic, and b) different times of the year. For example, is the thin ice in the Canada basin after FB assimilation a systematic feature throughout the 2018-2020 period? Or does this just occur in the one snapshot? I like the comparison to BGEP moorings as this shows a clear win for the FB assimilation at these locations. It would also be useful however to see e.g., monthly-mean spatial RMSE plots and time series comparisons (see some of my suggestions below). On this note, I'm also unsure why the authors have limited themselves to such a short period (2018-2020), when both CryoSat-2 and BGEP data are available back to 2010. I would strongly encourage the authors to extend their study to this full period in order to give more confidence that modelled thickness is indeed improved by assimilation of FB. I realise that this would create significantly more work and so may be an unrealistic request. Perhaps if some additional analysis shows convincingly that the assimilation is doing a good job between 2018-2020, then extending to 2010 will not be necessary.

In any case, I feel there is a fair bit of work needed before I can recommend this manuscript for publication. Therefore, I recommend major revisions for this article. My thanks to the authors for their work and I look forward to reading the next version!

*We thank the reviewer for their insightful comments and the work they put into the review. The method section will be rewritten and restructured taken your suggestions into account. In regards to the validation we feel the need to clarify that the AWI sea ice thickness should not be see as data set use to verify the quality of the SIT resulting from the assimilation. The comparison is included to illustrate the difference between the classical approach used to generate the AWI CS2 SIT and the assimilated SIT, as both data sets initially used the same freeboard data. Both data sets are than*

*verified against the BGEP mooring data. As this was not clear to the reviewer after reading the paper, we see the need to clarify this in the revised version of the paper.*
*Further we agree with the reviewer that validation in other regions than the Beaufort gyre is needed. The lack of validation in different areas of the Arctic will be addressed in the revised version of the paper. For this validation recently released ice mass balance boy data from the MOSAiC will be used in similar fashion as the BGEP data in figure 8.*
*In regards to the reviewers comment that validation in other periods than the winter would be desirable we see the need to point out that the AIW SIT data only covers the month October to April the simultaneous comparison in figure 8 can not be done at other times of the year. However the SIT from the assimilation is verified also during the summers of 2019 and 2020 in figure 7.*
*As figure 7 shows that the assimilation improves the sea ice thickness at all 3 BGEP mooring locations through out the full 2 year period displayed, we will not run the assimilation for the entire CryoSat2 period, because we do not believe that this will add more value and would be very costly.*

*Major Changes implemented:*
*   *Rewriting the Method section.*
*   *Rerunning the assimilation with the product specified errors.*
*   *Including an additional observation data set for validation.*
*   *Rewriting the result, discussion and conclusion to include changes of the two above points.*

General Comments:

Introduction
• The authors have done a good job at summarising the various uncertainties/assumptions related to deriving sea ice thickness estimates (L26-86), however one key piece of missing information is the choice of retracking algorithm. The roughness characteristics of the sea ice cause different degrees of scattering of the radar echo, which are then convolved to produce an average height of the snow-ice interface. A retracking algorithm which does not account for changes in scattering due to roughness may therefore produce a freeboard which is too high when sea ice roughness is high, and vice versa. Landy et al., 2019 for example have shown how the use of a 'physically-based' retracker can help mitigate these effects. I think the introduction section here should include a few sentences to highlight this as a source of uncertainty in sea ice thickness estimates.

*A discussion of the freeboard errors introduced by retrackers was added in line of the revised 56 manuscript.*

• L81: I'm a bit wary of saying that by assimilating FB, the effects of snow thickness and density errors are eliminated. Sea ice radar FB assumes that the radar echo is returned from the snow-ice interface, and this generally is not the case (e.g., Willatt et al., 2011; Nab et al., 2023). To appropriately model the scattering surface of the radar echo (and hence reduce uncertainty in FB) we need to account for snow thickness, density and other dieletric properties of the snow. Maybe just worth highlighting this as another source of uncertainty in satellite-derived sea ice thickness.

*The discussion of the uncertainties was rewritten in the revised version of the paper.*

• Figure 1 (and others throughout the manuscript): I suggest changing colours from red and green (in this case, the BGEP locations) to something more colour-blind friendly.

*The color scheme of all figure was revised for all figures.*

Methods and data

• I find section 2.2 a little hard to follow and am also struggling to relate it to section 2.6.
Is it essential to have these as separate sections? Can section 2.2 not be merged in with
section 2.6? In any case, it would be useful to provide more details about the various
model runs and how they were initialised etc, and potentially updating figure 2 with
more information. For example, what are 'VAR' and 'VARI'? I will summarise what I think
I understand, and please correct me if I'm wrong:
o An initial experiment was run between 1995-2020. This experiment was run as
an 80-member ensemble in coupled ice-ocean mode, and forced by ERA5
atmospheric reanalysis

*It is correct that the initial experiment was run between 1995-2020 and forced by ERA5*
*atmospheric reanalysis. The ensemble is static, meaning, that is was not run. The model error*
*covariance matrix was instead calculated from a historical run. Further descriptions are added in*
*section 2.2.1 in the revised manuscript.*

§ What were the initial ice/ocean conditions for this run?
§ How do you e.g., perturb the ice/ocean model parameters to create the
ensemble?
§ I'm not sure what is meant by increasing the variance to "account for
biases" (L138). Are you not just increasing the variance to prevent
ensemble collapse? Ultimately, you're hoping that the data assimilation
itself will reduce the biases
We use an static ensemble to reduce the computational costs. Normally the model error is determind
based on the esemble members covarriance matrix. In this study the covariiance matrix is calculate
based on model states from past dates in the initial run. For this 80 days are picked from the years
2010-2020 from the initial run. The mean feald is calculated from this model states and suptractd
from the current model state. This
o The 2018-2020 period of the initial experiment corresponds to the refRun
o The initial experiment at 2018-01-01 was used as initial conditions for both the
sicRun and the fbRun. Assimilation over the 2018-2020 period is performed
every 7 days.
o The increments from the assimilation runs are then saved, and then you
effectively re-run the sicRun and fbRun experiments over the 2018-2020 period,
except that the previously saved increments are now updating the model at
every time step (through linear interp of the increments from 7 days to 600 sec).
§ This is to prevent model shock after each assimilation cycle?
o I'm not sure where the 'static ensemble' fits into all of this? Could you explain?
More generally, could you explain the motivation for only focusing on the 2018-2020
period rather than the entire CryoSat-2 period (2010-present)? By utilising the entire
record I feel that you would be able to derive more rigorous statistics related to the
improved FB and thickness from assimilation. For example, time series comparisons of
monthly-mean FB and thickness between AWI and the assimilation run over the 2010-
present period, for different Arctic regions.

*The method section was completely rewritten in the revised version of the manuscript. Pleas find it*
*in section 2.2.*

L135: Could you provide more details on why you choose observational error estimates
of 15% for SIC and 0.15m for FB, given that on L175 and L166 you state that the

observational uncertainties are 10% and <0.07m, respectively?

*In the revised version of the paper the local product associated error estimate was used.*

L150-155: Is there any post-processing applied after assimilation to ensure that the updated SIC is bounded between 0 and 1? If so, how is this bounding applied? Particularly to the category terms.

*A description was added in line 217.*

I believe currently in the fbRun you are updating SIC and FB sequentially. Out of curiosity, is sea ice thickness updated during the assimilation of SIC, and similarly is SIC updated in the assimilation of FB? Do you also expect your results to differ if you first assimilate FB and then SIC?

*The assimilation of FB is done after the SIC assimilation because the limits of SIC determines where to assimilate FB. Assimilating first FB and than SIC might lead to differences, but we assume them to be minor especially since the assimilation is distributed over all time steps.*

Results

• L269-270: Suggest clarifying here that by "assimilation period" you mean November-March, as opposed to the whole 2018-2020 period.

*The mentioned part was rewritten.*

• L287: Does "beginning of October" and "end of winter" refer to a single day? Or a weekly average? Please clarify.

*This Line was rewritten.*

• Figure 5: Suggest including either correlation or R 2 values for both refRun and fbRun on each plot to quantify the improvement. I also suggest making this a 4-panel figure, and show the equivalent density plots for freeboard as well.

*R2 values and biases were added in table 1 for all month and both for FB and sea ice thickness.*

• L288: Can you also speculate why the "thin bias" problem is not improved with the assimilation of FB? Is this because the FB increment is spread linearly across the ITD categories (Equation 1), whereas in reality more weight should be given to the thinner categories?

*We don not believe that the thin bias originates from the ITD method. The reasons fort the this bias might originate from different sources depending on the region. This sources could be the modeled FB itself (this is in depth discussed in the revised version of the manuscript), the modeled sea ice density being overall lower, or differences in snow thickness.*

• Figure 6: I don't think panels C) and D) are all that informative. Especially considering that our observational estimates of snow thickness and ice density are inherently wrong. I suggest replacing these panels with freeboard comparisons (AWI-refRun) and (AWI-fbRun). It would also be good on each panel to include a mean RMSE value, as it

currently looks as if panels A) and B) might actually be similar in terms of RMSE.

*Figure 6 was not included in the revised version of the manuscript.*

• Figure 8: Could you speculate why the differences between AWI and fbRun are significantly large at the beginning of the 2018-2019 period? Is this a spin-up issue?

*We do not see evidence that this is a spin up issue. Comparing the improvement of the assimilated draft and the reference runs draft in figure 7 shows clearly that the draft in winter season 2018/2019 differs on similar magnitudes from the reference run as in the following years. In the revised version of the manuscript we changed the dots to lines and now this is more evident from figure 7.*

Discussion

• L400-410: Is the imprint of the FYI/MYI zone on the difference plot not just due to the fact that the AWI snow thickness and ice densities are assigned constant values depending on whether FYI or MYI? In your case you've replaced the density with a linear function after Mallett et al., 2020, but ultimately this imprint of a FYI/MYI mask is just a reflection of what the observations are (hence my suggestion above to remove).

*We removed figure 6 C and D all together.*

• L412: The panels in Figure 6 are errors in modelled sea ice thickness relative to AWI, not uncertainties in AWI sea ice thickness, right?

*Yes, this is a typo and was corrected.*

Minor suggestions:

*All minor suggestions were corrected in the reviewed version. We thank the editor for their work!*

Abstract

L4: Define SIC and FB acronyms

*The acronym are defined in line one and four.*

L6: Define AWI acronym

*AWI is defined in line 10.*

L10-11: This last sentence is rather vague. I suggest something like "Modelled sea ice draft errors are in good accordance with that of CryoSat-2 errors at BGEP mooring locations, with mean error differences less than 3 cm over the 2018-2020 period."

*Line 10-11 are not included in the revised manuscript.*

Introduction

L17: I'm not sure what is meant here by "need to affect the model variable that the assimilation aims to improve". Does it mean that we need to use an appropriate observational operator to

map the model state variable which is being updated, to the space of the observations? E.g., an operator to map thickness to freeboard?

*Line 17 has been rewritten.*

L18: I would clarify here that this statement relates specifically to Arctic sea ice predictability on seasonal-to-interannual timescales.
*This was corrected line 23 in the revised manuscript.*

L24: Suggest stating explicitly that initialising thickness is better for predicting Arctic sea ice on seasonal time scales or longer (sea ice area persistence is more important at short lead times).

*According correction are made in line 24.*

L55-57: Would also make reference here to the recently developed SnowModel-LG (Liston et al., 2020; Stroeve et al., 2020), which is being adopted in sea ice thickness products from e.g., Landy et al., 2022.

*Suggestion added in line 66.*

L65: The phrasing "add up" here feels a little vague. Does this mean that the error in sea ice thickness is equal to a linear sum of the errors in snow/ice/water density and FB?

*This line was edited out in the revised manuscript.*

L71: Suggest rephrasing to "The OSI-SAF ice type product (Aaboe et al., 2021) is one observational data set which aims to distinguish between FYI, MYI and ambiguous ice types."

*A more details discussion of the ice type data was added in the introduction.*

L76-77: The wording is a bit confusing here. Does it mean that the errors are systematically overestimated in the MYI zone and underestimated in the FYI zone in the OSI-SAF product?

*This line was edited out in the revised manuscript.*

L77-78: As far as I'm aware, sea ice area isn't required to generate thickness anyway, so this sentence seems redundant. Could you explain what you mean here? Unless you're referring to CryoSat-2-derived sea ice volume?

*This refers to the error of the ice type product, the sea ice area of FYI and MYI. A more details discussion of the ice type data was added in the introduction.*

Methods and data

L96: First time using PDAF acronym, please define.

*The acronym is introduced before it's first use in line 172 in the revised manuscript.*

L99: Suggest rephrasing to "An increment is the amount of change in a model state variable after the assimilation of observational data."

*This line was edited out in the revised manuscript.*

L106: First time using WMO acronym, please define.

L109: Suggest removing first sentence on L109 and changing L111 to "the key variables are snow thickness (h s ), snow density (rho s ), sea ice density (rho i ), and ocean water density (rho w )."
Or similarly just defining the terms explicitly on L109.

*This line was edited out in the revised manuscript.*

L142: 80 model states? Or 80 ensemble members?

*A clarification was added in section 2.2.1 of the revised manuscript.*

Results

L299: Suggest clarifying that the date 03-30-2020 is actually a 7-day mean.

*This line was edited out in the revised manuscript.*

L315: change to "independent of the satellite-derived FB data".

*Edited in the revised manuscript, now line 360.*

L330: change here and elsewhere in manuscript from "data is" to "data are"

*This line was edited out. Data is was changed in the relevant places in the revised version of the manuscript.*

Discussion

L410: I'm not sure on Crysophere reference guidelines, but you might need to change "Sievers et. al (in preparation)" to something like "Future work will analyse the effects of different variables..."

*Edited in the revised version of the paper in line 523.*

Reviewer 2:

Review of "Assimilating CryoSat-2 freeboard to improve Arctic sea ice thickness estimates" by Sievers et al. (2022)

This paper describes the assimilation of sea ice freeboard observations into a coupled ocean and sea ice model. The authors have compared the results of an analysis assimilating freeboard and sea ice concentration observations with a control run, and a run assimilating only sea ice concentration. They have used the AWI weekly CryoSat-2 sea ice thickness product, which is derived from the same freeboard observations as have been assimilated, and independent data from BGEP upward-looking sonar observations to validate their results.

*We thank the reviewer for their detailed comments, insightful suggestions and effort spent to assess our study.*

*Major Changes implemented in the revision:*
- *Rewriting the Method section.*
- *Rerunning the assimilation with the product specified errors.*
- *Including an additional observation data set for validation.*
- *Rewriting the result, discussion and conclusion to include changes of the two above points.*

General comments

The authors have developed a new method for the assimilation of FB (freeboard) observations rather than SIT (sea ice thickness) in their model. The technical method has clearly been well thought through and competently implemented. However, one of the main motivations for assimilating FB is being able to more easily quantify the associated observation uncertainties. Here, the authors have used a constant FB uncertainty, citing technical issues, which unfortunately means that they are unable to demonstrate the potential benefits of FB over SIT assimilation. Further, they have chosen to compare their FB assimilation results with the AWI SIT product, when a second run assimilating SIT into their own system would have provided a much improved comparison. A comparison to the AWI data is of interest, but is unable to adequately demonstrate the benefits of assimilating FB over SIT. These flaws in the methodology unfortunately mean the impact of the paper as it stands is limited.

*We agree that the constant error estimates are a major short come of our study and have include reruns with variable error estimates in the revised version of the paper.*
*We also see the point of the reviewer that assimilating SIT in comparison would strengthen our point. This would however only add value if the error estimates of the SIT product would be as reliable as the FB error estimates. We have serious doups that this is the case and take this as one more motivation to assimilate FB instead of SIT. A more detailed discussion of this was added in the introduction: Error estimates of SIT products are difficult for many reasons. The variables used for snow and ice density and snow thickness are based on climatologies and point measurements taken during a time where the state of the Arcitc sea ice was significantly different. But the main reasons we argue that the error estimate of the SIT product is unrealistic are the ice type discrimination. A discussion of ice type data sets has long been lacking. Ye et al 2023 has recently assessed different sea ice type products including the OSISAF ice type product used in the AWI CryoSat2 data product. Comparing ice type data to NSIDC sea ice age data (Tschudi et al., 2020) they find that OSISAF ice type data for FYI has a bias of 0.42-0.6 $10^6 km^2$ and for MYI of -0.54 – -0.35 $10^6 km^2$. The comparison of Ye et al 2023 only includes FYI and MYI area and compares it to satellite obtained ice age products, not ambiguous areas. NASAs ice-chart based sea ice type product G10033-V001 shows that the ambiguous area is significantly lager than the one accounted for in the OSISAF product, or any of the ice type products compared in Ye et. al. 2023. Errors from ice type are not accounted for in the SIT error estimate at all.*
*Apart from the issue that the error of the SIT is most likely underestimated in places where the ice type is ambiguous the SIT errors are in the order of up to 40% of the SIT while the FB errors are in the order of up to 14% of the FB values. This is why we decided to compare the resulting SIT to the AWI SIT and independent measurements. Insito observations in the Arcit are however hard to come by. An additional in sito SIT data set was used to compare both the freeboard assimilated SIT from the model and the AWI CryoSat2 SIT to. A detailed discussion is included in the revised manuscript.*

The description of some of the methods needs further clarification (see comments below). The results section needs to include more of a critical assessment - a description of the results is given, but why might we be seeing these? Some insight appears in the discussion section, but to avoid the reader having to keep referring back to the earlier figures, it should be included and expanded on in the results section instead. A discussion section can then feature more general points to tie the paper together. This would also remove some of the repetition of earlier information that appears in the discussion section. Additionally, the conclusions presented are inconsistent and some of the statements require further justification (see comments below).

*The methods was restructured and rewritten. We acknowledge that the reviewer prefers a mixed result and discussion section, and have adjusted the revised version of the manuscript where we could see that it would improve the text.*

Additionally, the paper would benefit from more discussion of the theory behind why we might expect the assimilation of FB to be an improvement over SIT, see e.g. Kaminski et al. (2018) as a starting point: Kaminski, T., Kauker, F., Toudal Pedersen, L., Voßbeck, M., Haak, H., Niederdrenk, L., Hendricks, S., Ricker, R., Karcher, M., Eicken, H., and Gråbak, O.: Arctic Mission Benefit Analysis: impact of sea ice thickness, freeboard, and snow depth products on sea ice forecast performance, The Cryosphere, 12, 2569–2594, https://doi.org/10.5194/tc-12-2569-2018, 2018.

*The suggested study was is mentioned in the revised version of the paper in line 106-107.*

More generally, as detailed below, there are some confusing and contradictory statements in places. There are also some missing citations, statements that need to be quantified, and confusing notation is used in figures and equations.

*Extra attention was payed to the mentioned issues in the writing process of the revised manuscript.*

Specific comments

Major comments:

Line 135: Elaborate on how the values for the SIC and FB observation errors were selected. As stated above, it seems strange to use a constant FB error, given the arguments for using FB over SIT.

*The revised version of the paper included reruns with variable error estimates.*

Line 166: Uncertainty in the FB dataset is given as 0-0.07 m, but the authors have chosen to use 0.15 m as the observation error. This needs further explanation here.
Line 175: Uncertainty in the SIC data is given as 10%, but the authors are using 15% as the (constant) observation error. This needs further explanation.

*The revised version of the paper included reruns with variable error estimates.*

Line 196, Figure 7: Why not average the 10-second data first to get a daily mean, and then calculate the difference to the model daily field? And actually, this contradicts the caption of Figure 7. This needs clarification.

*The STD in figure 6 (former figure 7) is calculated based on the 10-second data to display the variability of the BGEP data. The mean to calculate this std is also shown in figure 6 for consistency. More details can be found in section 2.5 of the revised manuscript.*

Line 212: I think the authors are describing an incremental analysis update (IAU) method, suggest describing as this as such (citation Bloom et al. (1996): Bloom, S. C., Takacs, L. L., da Silva, A. M., and Ledvina, D.: Data assimilation using incremental analysis updates, Mon. Weather Rev., 124, 1256–1271, 1996.)

*This was correct in the revised version (line 205 in the revised manuscript).*

Line 214: The authors say here that FB is converted to SIT before subtracting the increment from the model. This implies that the conversion is performed for the FB observations, and thus that the authors are actually assimilating SIT. However, from reading the method in the paper, it seems that FB increments are in fact produced, and applied to the model radar FB (which is the novel part of the paper) before this is converted back to model SIT in order to distribute the increment over the thickness categories. Therefore, this statement needs to be reworded as it's rather misleading. Also line 230: "Since FB is not a model state variable, it needs to be transformed into sea ice thickness before it can be treated..." needs explanation added along the lines of "Since FB is not a model state variable, model SIT needs to be transformed into FB before the FB increment can be applied..." etc.

*The method section was completely rewritten in the revised manuscript a described of the FB to thickness conversion is discussed in section 2.2.2*

Line 227-230: Citation needed for the statement about FB measurement reliability in regions of low SIC. How were the values of 80% and 0.05 cm chosen?

*In the revised version  this is discussed in line 220.*

Line 238-241: The description of how the work of Mallett et al. (2020) relates to the method used by the authors is confusing and needs clarification. For example, is it the calculation of c_s in equation 3 that uses a seasonal snow density? If the same snow density is used throughout, why mention the linear function? Additionally, 10 cm is quoted on line 60, and 15 cm on line 239, for the improvement in SIT from the method of Mallett et al. (2020).

*This was clearifyed in the revised version in section 2.1.*

Line 269: Why might the RMSE increase?

*The discussion of the RMSE increase was moved to the result section.*

Line 272: Why? And could this indicate an issue with the assimilation?

*This is discussed in the revised version of the paper in line 305 and 320.*

Line 284: "differences can illustrate the impact of changing the method of converting FB to ice thickness". Since there are model uncertainties for the runs and observation uncertainties in the AWI data, this is not a clean comparison of the impact of assimilating FB over SIT. A better choice would have been to compare a run assimilating FB with a run

assimilating SIT. As discussed, the AWI data has different characteristics in the snow thickness and sea ice density, so the comparison is not an assessment of the benefits (or otherwise) of assimilating FB over SIT.

*The aim of comparing the two SITs is not to asses if FB assimilation is better than SIT assimilation, but to show that the FB assimilation gives SIT results in a comparable range as the direct conversion. With the inclusion of the MOSAiC observations this is more evident.*

Section 3.2: Why might we be seeing these differences in results?
Section 3.3: What do the results indicate?

*The discussion and result section were rewritten in the revised manuscript*

Line 340-341: Should show mean difference in Table 1 here too, especially as bias is discussed

*Biases and correlation coefficients were added for all month and for both FB and sea ice thickness in table 1 in the revised manuscript.*

Line 360: Why does SIC improve?

*Because it's assimilated. The answer seems obvious. Pleas elaborate if it was not answered in the revised manuscript.*

Line 369: What about 2019-2020?

*The fbRuns sea ice thickness is in all points closer to observations than the sicRuns sea ice thickness. Also in 2019-2020 (figure 6 in the revised version of the manuscript).*

Line 372: Quantify this, show mean difference and RMSE.

*RMSEs were added in table 3 and a one sided t-test was performed to test statistical significant where relevant.*

Line 378: Why might the assimilation run be worse than the reference run?

*This is discussed in the revised version of the paper in line 305 and 320.*

Line 397: Would the "week following the 30th March" be better described as April? It is stated earlier in the paper that only FB observations between November and March are assimilated due to melt pond issues. Why was this week chosen?

*The weeks were chosen because they are the first and last week of the assimilation season. As the last week starts with March 31 it is included.*

Line 401: Would be helpful to show an ice type figure as this is referred to in the discussion. Needs more information on how ice type is used in the assimilation runs and the AWI product for interpretation of the differences described.

*The ice type is not used in the assimilation. The assimilation uses model values instead of constant values used by the AWI sea ice product. This model values are described in the*

*method section and are depending on the amount of salt in the sea ice and on the snow fall in the forcing. Figure 6 was exuded in the revised version of the manuscript since it seems to add more confusion than value.*

Line 412-415: Figure 6 shows differences between the AWI product and the assimilation runs, rather than specifically uncertainties in the AWI dataset. This part is a bit confusing. Why is this a prerequisite for comparison to BGEP?

*Figure 6 was excluded in the revised manuscript.*

Line 418, 420-422, 440, abstract: Contradiction in conclusions - e.g. in Section 3.3 it is stated that the fbRun and AWI results are "almost equal" and there's "no clear bias", but lines 420-422 state that FB assimilation leads to improvement. Suggest calculating statistical significance of the mean and RMS differences to determine a definitive and justifiable conclusion, and be consistent with this throughout paper. Additionally, line 444: "comparable SIT as the AWI SIT product", but e.g. figures 5, 6 show differently... (also line 453)

*Not all improvements compared the AWI data to the assimilated data. In some line mentioned by the reviewer it is meant that the assimilation improved the SIT compared to the non assimilated run. This was clearifyed in the revised version of the paper. We also added more SIT in sito observations from MOSAiC to compare to both the AWI SIT and the assimilated SIT to strengthen our point.*

Line 452: Need to present the aims of the study in the introduction to the paper rather than just at the end.

*A clear statement about the aim of the paper was added in the introduction.*

Line 451: Being able to use FB uncertainty would really improve the paper, can these technical issues be overcome?

*The technical issue was overcome and the revised version includes the variable error estimate.*

Line 453: If the conclusion is that assimilating FB gives similar results to SIT, then what is the justification for implementing a new method?

*The revised version of the manuscript shows that the assimilated sea ice thickness is better than the classical derived sea ice thickness. It also demonstrates where the strength of the FB assimilation lays (line 422-431 for example).*

Minor comments:

Line 23: Add an extra sentence on what is meant by SIT and SIC having a memory
*Explanation added in line 30.*

Line 26: SIT data is also available from other satellites, need citation for CryoSat-2 being the "most commonly used"

*Has been changed to "The longest FB observations from a satellite with a polar orbit " in the revised manuscript.*

Line 31: Add "to reduce the uncertainty in the observations" to the end of this sentence for clarity. Also add explanation of why the uncertainties when the model converts back to SIT are less important (i.e. because they don't affect the assimilation).

*Which observations are meant here? Our point is not that the uncertainties effect the sea ice thickness less, when they are model derived, but that the uncertainties of the sea ice thickness used in the Kalman filter is effected by significantly more sources than the FB error. The relevant part in the discussion was edited and should now be more clear.*

Line 35: "to a large extent" - quantify this

*This sentence was edited out in the revised manuscript.*

Line 46: "most products" - citations needed

*Citation added.*

Line 65: Inconsistent - have just said the influence of sea water density is negligible

*Since the ocean model calculates the water density and the water density is part of equation 2 it would be unnecessary to use a newly introduced constant value.*

Line 70: OSI SAF ice type data is not the only product available. Is used in this paper what is meant? Or for CryoSat-2 products? Clarify.

*The AWI data set uses the OSISAF ice type data and it is the most commonly used ice type data set in the CryoSat2 sea ice thickness products (Sallila, H. et. al. 2019). In the revised version a discussion of the accuracy of ice type data in general will be included in the introduction.*

Line 77: "unambiguous" should be "ambiguous"

*This sentence was edited out in the revised manuscript.*

Line 95: sea ice thickness and freeboard datasets

*This sentence was edited out in the revised manuscript.*

Line 103: "more recent versions" - need to be more specific here, version numbers, citations.

*Version are state in the beginning of the paragraph (line 119).*

Line 105: "Icepack" needs a citation, and "by default linked" needs clarification
*Citation added in line 124-125 in the revised manuscript.*

Line 109-111: The shorthand "rho_i" etc are mentioned before the explanation of the variables on line 111.

*The method section was rewritten in the revised version of the manuscript.*

Line 116: Suggest rewording, e.g.: "The density of fresh ice is set to 882 kg/m3 and the amount and density of brine are calculated to produce an estimate of ice density. The sea surface water density is calculated in NEMO." The detail of which functions were used is not needed.

*The method section was rewritten in the revised version of the manuscript.*

Line 119: Not just the area in red? The blue part as well?

*Clarification was added that the blue and orange part covers large areas of the model domain and there by the red area.*

Figure 1: The coastline in the zoomed area does not match up (can see it's rotated, but it still is not the same area covered)

*The same projection was used in the revised version of the manuscript.*

Figure 1 caption: Date is needed for the CryoSat-2 data shown.

*The date was added in the revised manuscript.*

Line 137: What is meant by "static ensemble"?

*he method section was rewritten in the revised version of the manuscript.*

Line 138: How was the run analysed? What is the reference used to assess model biases? Needs more information.

*It was compared to OSISAF. This is now mentioned in line 191.*

Line 160: Define "DTU21"

*DTU21 is the name of the sea surface model not an acronym.*

Line 166-167, also line 176: What kind of interpolation? Bilinear? etc. Probably don't need to specify CDO was used.

*Interpolation was specified in the revised version of the paper.*

Line 167-168: Define "per one assimilation time" - one week? Suggest referring to "orange lines" (or tracks) in Figure 1 rather than "orange area"

*Changes can be found now in line 238.*

Line 171: SSMIS is the instrument, the satellite is the DMSP series
*This was edited in the revised version.*

Line 174: Is this definitely Level 4 data? The 10 km daily product (OSI-401-b) is Level 3, which makes more sense for assimilation. I don't think there's a 10 km Level 4 product?

*We changed the OSAISAF product to the 25x25km climate data record, because it comes with better error estimates for our purposes (personal communication with Fabrizio Baordo).*

Line 179: "sea ice thickness output from the assimilation" should presumably be "sea ice thickness model output from the assimilation run" or similar, unless it's output specifically from the assimilation step only

*This line was edited out in the revised manuscript.*

Line 183: Need a citation for NSIDC AMSR2 snow depth product

*No explicit citation is added in the product description of the sea ice thickness. An other study (https://doi.org/10.5194/tc-2023-40) citing the product cited the AWI sea ice thickness product describe here. Hence we added the same.*

Line 184: "following Mallett et al. (2020)" needs more information, add e.g. "as described in section..."

*reference to relevant equation in mehtod section was added.*

Line 188: Remove the word "completely" as it's not an independent dataset as it is derived from the same freeboard observations - which is the reason it was chosen presumably?

*"completely" was removed from the sentence.*

Line 189: Suggest replacing "based on" with "from the model, adjusted using" for clarity

*Not relevant in revised text.*

Line 194: Expand on what is meant by "tilting errors"

*Explanation added in line 270 in revised text.*

Line 200: Are both SIC and SIT weekly assimilation?

*Yes*

Line 201: SIC observations are available over the summer months, so was this done to aid comparison? Clarify.

*Studies have found that the summer melt ponds lead to underestimated SIC in satellite passive microwave measurements. More detailes and sources are listed in line 255 in the manuscript.*

Line 220: I think data from the restart file should be described as data at t=0 (or t_0 as in figure 2). What is inc_icon?
*Figure 2 and line 220 were edited in the revised manuscript.*

Equation 1: What about t_n (as in figure 2)?

*Not relevant in revised text.*

Figure 2: Equation 1 and Figure 2 should use consistent notation, and these should all be defined (e.g. VAR, VARI, cat are not defined)

*Not relevant in revised text.*

Equation 2: Need a citation for this equation. What is r_0? Radar freeboard at t=0?

*Changes can be found in line 144 in the revised manuscript.*

Line 237: Repetition of line 234, combine this information.

*Not relevant in revised text.*

Line 255: Detail of function names is not needed, suggest reword to "After each assimilation, sea ice thickness in all categories is set within the range defined in Section 2.1."

*Not relevant in revised text.*

Equations: Notation is quite confusing, especially use of subscripts such as r, 0, which are not clearly defined.

*The revised manuscript was double checked for missed definition of variables. To our best knowledge no undefined variables should remain.*

Line 261: Isn't the orange area just an example track for a specific week? Reword this.

*It was specified that the orange tracks are an example.*

Figure 3: Is y-axis ice concentration (fraction) or ice area? Would also be clearer shown as a line graph rather than plotted as points.

*Axis was renamed in the revised manuscript.*

Figure 4 (top panel): Clarify that the green SIC is underneath the red (and not the blue, or missing)

*Clarification was added.*

Figure 5: What are the lines on the outer right edges and tops of the plots? What is meant by "initially observed FB locations"? Also add the years to March and October.

*The lines show univariate kernal density estimate. This is specified in the caption in the revised manuscript. Also were all years used in the revised manuscript.*

Line 308: Better to use e.g. "notably" instead of "significantly" as implies statistical significance in this context.

*Statistical significance was determined where relevant.*

Line 321: Does this mean there is no interpolation of the model data, just using the closest grid point to the observation location?

*Yes. The modeled sea ice thickness is an average over the 10x10 km grid. Hence it makes sens to only determine which grid point would cover the the BGEP location. This is specified in section 2.5 in the revised manuscript.*

Line 325: Days rather than "time steps"?

*Not relevant in revised text.*

Line 329: "indicates periods with no ice present" in the observations

*Changes added in line 377 of the revised manuscript.*

Line 330: Explain why no data are available Figure 7 caption: Method should be in the text (and contradictory, see comment for line 196 above). What is the lighter/darker shading on the figure?

*Method is described in section 2.5 in the Method section of the revised manuscript.*

Line 333: Statistically significant?

*Statistical significance was determined with a one sided t-test where relevant.*

Line 349: Suggest changing "assimilated SIC and FB and the..." to "assimilated SIC and FB observations and the..." for clarity.

*Not relevant in revised text.*

Line 366: Can the model change SIT/add new ice?

*When no ice exists and the SIC suggest formation of new ice 10 cm new ice is added in the thinnest category as described in section 2.6.1. FB is only assimilated where SIC>80%, so no new ice is formed here.*

Figure 8: Suggest choosing a different (non-filled) symbol for the fbRun as not possible to see the '+' symbol when the points are overlapping. Extend the x-axis as the symbols are cut off at the edge

*Figure 8 was recised.*

Line 380-382: Confusing wording here.

*Not relevant in revised text.*

Line 387: Is this mean sea ice thicknesses?

*It would rather be the mode sea ice thickness*

Line 389: Would be better if lines were overlaid to show this

*Not relevant in revised text.*

Lines 389, 390: I think "dominate" should be "dominant", but what does this mean?

*Not relevant in revised text.*

Line 407: Filtered how?

*Not relevant in revised text.*

Line 416: "assimilated draft" implies this data has been assimilated, suggest changing to "draft from the assimilation runs" or similar. Also line 417, "assimilated data" should be "assimilation run data", or similar, and also line 419.

*Not relevant in revised text.*

Line 417: Suggest "time steps" should just be "times".

*Not relevant in revised text.*

Line 460: Figure 8 results are at the BGEP locations only (add this information to sentence)

*Not relevant in revised text.*

Technical corrections

*We thank the reviewer for the corrections and apologize for the missed errors. We will proofread the revised version one extra time.*

The paper could do with an edit to improve readability Acronyms in the abstract need to be defined (or not used)

Line 34: "climatology's" should be "climatologies"

Line 60: "Resent" should be "Recent"

Line 92: "on an assimilation method" should be "of an assimilation method"

Line 101: "assimulation" should be "assimilation", also "Nemo" should be "NEMO"

Line 129: "nor" should be "or"

Line 137: "from a initial" should be "from an initial"

Line 142: "where" should be "were"

Line 165: "where" should be "when"

Line 185: Suggest replacing "with the help of" with "using" (and similar elsewhere)

Line 190: Suggest changing "to validate against" to "to validate the model against"

Line 195 (and 320, 322, 328): "standard derivation" should be "standard deviation"

Line 211, 224: Write "not zero" rather than "!=0" in the text

Line 212: "was" should be "is"

Line 214: "it's" should be "its"

Line 217: I think "fractal" should be "fractional"

Line 218: "here after" should be "hereafter"

Line 224: "exist" should be "exists"

Line 258: "Consentration" should be "Concentration"

Line 259: Define "RMSE" acronym line 281: "comparable" should be "comparably"

Figure 3 caption: "location" should be locations"

Line 333: "in-sito" should be "in-situ"

Line 390: repetition of "the"

Line 424: "3 years assimilation run is" should be "3-year assimilation run are"

Lines 425-426: Suggest rewording: "The presented method calculates an increment using modelled FB and then converts the..."

Line 446: "does" should be "the"

Line 449: "spacial" should be "spatial"

*Line 454: "recommendable" should be "recommended"*

---

## Author Response (AR2)

*We thank the reviewer for their responds and comments. Pleas find all answers below the comments. The revised manuscript also includes minor spelling corrections.*

**L37 - not sure what is meant by "sea ice is not directly measurable" suggest removing**

*Authors Response: removed in reviewed manuscript.*

**L111 - The statement "FB error is more accurate than sea ice thickness error" is unclear - do you mean the error is better constrained?**

*Authors Response: Yes better constrained is the correct term missing.*

**L121 - change to "how does the modeled sea ice thickness after assimilation of FB compare to SIT from a conventional…"**

*Authors Response: changed in the revised manuscript.*

**L209 - change to "ensemble of model forecasts/simulations"**

*Authors Response:changed in the revised manuscript.*

**Figure 2: I don't think Figure 2 provides any additional information to what is in the text. I suggest removing. In any case, Figure 2 says "3 months" around assimilation date, while L250 says "2 months".**
**Needs changing.**

*Authors Response: We corrected the figure text, but decided to include the figure as a visual aid.*

**L358-L360: reference to figure 6 & 7 comes before figures 3,4,5. Suggest reordering so figures are referenced in order.**
**L406: same for figure 8**
**L428: same for figure 9**

*Authors Response: We removed the references to the figures in the method section.*

**L445 & 463: what is meant by "the physical balance" ?**

*Authors Response: A model is basically a set of physical differential equations, if extra mass or energy is added (as is done in assimilation) the physical balance will be reestablished over time.*

**L455-459: I'm not sure that the increase in RMSE over time is just due to the seasonal cycle of sea ice area. In fact, I would expect the RMSE in summer to potentially be largest compared to other seasons (maybe this could be checked if the refRun extends into the summer). As you say, winter SIC errors are primarily coming from the ice edge, as SIC poleward of the marginal ice zone is going to be pretty close to 100% in the model and reality. Meanwhile the local SIC variability in summer can be very large, which the model will not necessarily be able to capture accurately. Also, there will be some amount of error growth in the refRun associated with model drift. I think this is worth mentioning**

*Authors Response: We agree that the RMSE is largest in summer. However, only winter month are displayed (November to March). The area with large summer RMSE is in November already covered by close to 100% SIC. The largest errors accrue in the Atlantic region. Here the ice edge gets longer through out the winter season. Editions were made to the manuscript to clarify this.*
*A longer (15y) SIC analysis of the background run, from which the refRuns, sicRun and fbRun are started, show that the SIC model drift is neglectable. It was calculated in comparison to OSISAF SIC.*

**L472: suggest changing to "the assimilation is working as expected"**

*Authors Response: We changed the sentence to "Any differences between the two data sets therefore indicate the impact of the here introduced FB assimilation in contrast to the method of directly converting FB to sea ice thickness.". That the assimilation is working as expected is already shown in Figure 4.*

**L478-479: There will also be inherent differences between the two as DA computes a "best guess" based on observational and model uncertainty, while the observations are point**

**estimates.**

*Authors Response: That is correct. The point was to underline that two different methods were applied to the same data set and the aim is to compare the different results. The assimilation method is evaluated as a hole, including the calculation of the state estimate, not only from the point where the increment is added to the model. The paragraph was altered in the revised manuscript to underline this (including the changes due to the comment above).*

**Table 1: Do the correlation values correspond to the mean all grid point correlations?**

*Authors Response: For clarification the table text was changed to: "Monthly mean correlation coefficient and mean bias between the weekly AWI sea ice thickness (SIT) and FB and the fbRun SIT and FB for the entire assimilation period from 2018-01-01 to 2020-12-31. Only grid points covered by both the AWI FB data and the model were considered."*